# Certifying Capabilities from Finite Tests: When Is It Possible?

**Changlong Wu** [1]   **Jin Sima** [2]   **Wojciech Szpankowski** [2,3]

## Abstract

Modern foundation models are evaluated through broad capabilities such as arithmetic, reasoning, safety, and robustness, yet it remains unclear in a principled sense when finitely tests can meaningfully certify such claims. We develop a rigorous theory of capability evaluation by formalizing evaluation as inference over a task family and asking when guarantees over the full family can be inferred from a strict subset of tests. We analyze two canonical regimes. In stochastic multi-environment evaluation, we characterize when uniform certification is possible across multiple environments and show that the sample complexity is governed by a $\chi^2$-radius of the environment family, yielding near-optimal evaluation protocols with matching lower bounds under a natural overlap condition. In contrast, for worst-case, rule-like capabilities, we establish fundamental impossibility results. Even for structured model classes such as Boolean circuits of bounded size, black-box evaluation cannot, in general, certify global properties. Together, these results provide a principled framework for understanding when finite evaluation can and cannot certify capabilities.

## 1. Introduction

Modern machine learning systems are increasingly assessed by their broader *capabilities* rather than performance on a single task. For example, we ask whether a model can reliably carry out arithmetic reasoning, behave safely across diverse prompts, or generalize across distributions and task formats. Such questions are now central to how we evaluate large language models (LLMs) and other foundation models (Liang et al., 2022; Vendrow et al., 2025). Yet we still lack a clear theory for *when and how* a capability claim can be justified from a finite set of tests.

Most existing evaluations are benchmark-driven. Static test suites such as GLUE and SuperGLUE (Wang et al., 2018; 2019) were designed to measure broad progress, but large models quickly saturated them, often without commensurate gains in robustness or reliability (Vendrow et al., 2025). More recent efforts like BIG-bench (Srivastava et al., 2023) and HELM expand coverage to hundreds of tasks and metrics, but they remain finite collections with fixed test sets. In parallel, dynamic and adversarial efforts such as Dynabench (Kiela et al., 2021) highlight a recurring gap: strong performance on established benchmarks does not necessarily carry over to newly created or rare cases. The common thread is that benchmark scores summarize behavior on a limited set of instances, while capability claims are implicitly about a much larger family.

These developments raise a foundational question:

> *When can finite evaluation genuinely certify a model's capability, and when is such certification fundamentally impossible?*

This paper argues that addressing this question requires reframing evaluation as an *inference problem* rather than benchmark scoring. The key observation is that a *capability* is not tied to a fixed test set; instead, it should be defined over a *task family* together with an aggregation rule over outcomes. For example, arithmetic capability concerns correctness across the full set of relevant problems, while safety capability concerns behavior across a broad space of prompts and contexts. Evaluation is therefore the problem of inferring a *global property of a model over an entire task family* from only finitely many observed interactions.

We formalize this viewpoint in a general framework for *capability evaluation*. A task family $\mathcal{T}$ defines the scope of behavior we wish to certify. A model $f$ (viewed as a black-box response function or induced response process) interacts with tasks in $\mathcal{T}$, and a capability is a functional $\mathcal{C}_\mathcal{T}(f)$ of the model's induced responses over all tasks in $\mathcal{T}$. An evaluation protocol may interact with the model through arbitrary queries, possibly adaptive and not drawn from $\mathcal{T}$, and must infer $\mathcal{C}_\mathcal{T}(f)$ from the resulting responses. Within this framework, an evaluation is *generalizable* if it can accurately infer a capability defined over the full task family while making only finitely many queries.

[1]University of Arizona, Tucson, AZ, USA  [2]Purdue University, W. Lafayette, IN, USA [3]Jagiellonian University, Krakow, Poland. Correspondence to: Changlong Wu <clwu@arizona.edu>.

*Proceedings of the 43rd International Conference on Machine Learning*, Seoul, South Korea. PMLR 306, 2026. Copyright 2026 by the author(s).

Importantly, capabilities are defined *relative to the choice of task family*, and different choices of $\mathcal{T}$ capture fundamentally different evaluation goals. To avoid an overly broad narrative and to demonstrate how precise technical results can be obtained, we focus on two representative evaluation paradigms that expose sharply contrasting behaviors.

**Stochastic task families.** Perhaps the simplest and most widely used form of capability evaluation is to estimate a model's *expected* performance under a fixed data distribution using i.i.d. samples. In practice, however, models are deployed across domains and populations, so evaluation must often certify performance *across* multiple environments. This motivates our first evaluation setting. We are given an environment family $\mathcal{P} \subset \Delta(\mathcal{Z})$ and a fixed *stochastic* model $f$, whose response (and hence loss $\ell_f$) may be randomized. The capability of interest is the *risk profile*

$$\mu_p(f) := \mathbb{E}_{z \sim p}[\mathbb{E}[\ell_f(z) \mid z]], \qquad \forall p \in \mathcal{P},$$

where expectation is over both $p$ and $f$. We seek to estimate $\{\mu_p(f)\}_{p \in \mathcal{P}}$ *simultaneously* using only finitely many queries to $f$, with high-probability uniform guarantees over $p \in \mathcal{P}$. We show that without additional structure relating environments, a worst-case lower bound of $\Omega(m/\varepsilon^2)$ samples is unavoidable when $|\mathcal{P}| = m$. We then identify when evaluation becomes effective by introducing the $\chi^2$-*radius* $V(\mathcal{P})$, which quantifies how well $\mathcal{P}$ can be simultaneously covered by a single proposal distribution, and we give a protocol achieving $O\left(\frac{(1+V(\mathcal{P}))\log(m/\delta)}{\varepsilon^2}\right)$ samples for estimating the full risk profile. Finally, we establish near-tightness by showing that the dependence on $V(\mathcal{P})$ is unavoidable for natural evaluation strategies. Moreover, even when $V(\mathcal{P}) = O(1)$, estimating the *entire* profile still requires $\Omega((\log m)/\varepsilon^2)$ samples, even for fully adaptive protocols. We also extend the framework to infinite families $\mathcal{P}$ via covering arguments, yielding generalizable evaluation guarantees under metric structure.

**Combinatorial task families.** A key feature of the stochastic paradigm is that, because tasks are specified by distributions, generalization can be certified from i.i.d. sampling *without* relying on the internal structure of $f$. This changes qualitatively for *combinatorial* task families, where tasks are discrete and structured (e.g., arithmetic instances, logical constraints, or prompt templates) and the capability of interest is correctness over an entire combinatorial set. In this setting, a fixed model can concentrate all of its errors on a small, unqueried subset of tasks, making finite testing inherently fragile. We formalize this phenomenon by proving strong impossibility results showing that, for natural combinatorial task families and rich model classes (e.g., Boolean circuits of size $s$), no evaluation protocol using $T(n) = o(2^n/M(s))$ queries can reliably distinguish

perfect performance from performance with additive error $M(s)$, even when the number of queries exceeds the model's description length. These lower bounds highlight a fundamental limitation of benchmark-style certification for combinatorial reasoning and clarify that meaningful guarantees require additional structure *linking* the model and the task family.

**Contributions.** In summary, this paper makes three key contributions: (i) we develop a formal theory of *capability evaluation* that treats evaluation as inference over task families and characterizes when finite testing can *generalize* to certify a capability over an entire family; (ii) in the stochastic multi-environment regime, we give protocols and nearly matching lower bounds, showing that the sample complexity is controlled by an intrinsic overlap measure of the environment family (the $\chi^2$-radius); (iii) in the combinatorial regime, we establish fundamental impossibility results showing that without explicit structure linking the model and the task family, finite black-box evaluation cannot certify global correctness. Together, these results provide sharp characterizations of the information-theoretic limits of certifying capabilities from finitely many tests.

### 1.1. Related Work

A large body of work evaluates general-purpose models via aggregate performance on fixed task suites, from early multi-task benchmarks such as GLUE and SuperGLUE (Wang et al., 2018; 2019) to broader collections like BIG-bench and HELM (Srivastava et al., 2023; Liang et al., 2022). These benchmarks have been instrumental in tracking empirical progress, but their finiteness leaves implicit the inferential leap from performance on a particular suite to claims about a model's behavior over a broader task family. Dynamic and adversarial evaluation efforts such as Dynabench (Kiela et al., 2021) sharpen this concern by adaptively generating new test cases that surface previously hidden failure modes, underscoring the brittleness of static evaluation. Our work targets a more foundational question: *what, if anything, can finite evaluation certify about a model's behavior over an entire task family?* From a theoretical perspective, our multi-environment setting is closest in spirit to multi-mean (or multi-expectation) estimation, where shared samples are allocated to estimate many quantities efficiently (Elvira et al., 2015; Demange-Chryst et al., 2023). This line of work emphasizes variance reduction for *finite* distribution families, but it does not typically provide sharp high-probability query-complexity characterizations. Our impossibility result in the combinatorial regime is also philosophically aligned with recent work framing limits of backdoor detection (Pichler et al., 2024), as well as earlier works on formal verification of neural networks (Katz et al., 2017; Ehlers, 2017), though they concern different objects and fail-

ure modes. More broadly, a large body of work in theoretical computer science and information theory studies the sample complexity of testing and learning from unknown distributions, including identity, closeness, and independence testing (Canonne, 2020), as well as work deriving minimax rates and information-theoretic lower bounds under structural or communication constraints (Acharya et al., 2015; Canonne et al., 2018; Diakonikolas et al., 2014; Acharya et al., 2019). These results characterize the statistical difficulty of learning from unknown distributions, but they generally do not capture the structural coupling induced by evaluating many tasks on a single fixed model.

## 2. Problem Formulation

We formalize *capability evaluation* as inference over a specified *task family* $\mathcal{T}$. A capability is a claim about a model's performance across all tasks in $\mathcal{T}$. An evaluation protocol observes only finitely many interactions with the model, yet aims to infer or certify such a family-level claim.

### 2.1. Models

A *model* is the system under evaluation. We represent it by an element $f \in \mathcal{F}$, where $\mathcal{F}$ is a class of possible models (e.g., trained LLM checkpoints, neural networks, or Boolean circuits of bounded size). The model is fixed throughout evaluation: it cannot be retrained, modified, or inspected internally. The model may be deterministic or internally randomized, but each interaction induces a well-defined distribution over observable responses.

### 2.2. Items, tasks, raw outputs, and task responses

We distinguish between *items* and *tasks*. An *item* is a raw evaluation instance: a question, prompt, input-label pair, environment state, or tool-use scenario. Let $\mathcal{I}$ denote the item space. A *task* is a pair $t = (Q_t, \phi_t)$, where $Q_t \in \Delta(\mathcal{I})$ is a distribution over items and $\phi_t$ is an evaluation or judge map. To evaluate model $f$ on task $t$, an item $I \sim Q_t$ is drawn, the model produces a raw output $O_f(I)$, and the judge returns an *observable response*

$$L = \phi_t(I, O_f(I)) \in \mathcal{L}_t.$$

The response $L$ is the only quantity available to the evaluator. It may be binary (pass/fail), real-valued (loss or score), or categorical. For a given model $f$ and task $t$, repeated interactions induce a *task-response* distribution $P_t^f \in \Delta(\mathcal{L}_t)$ through the following sampling chain

$$I \sim Q_t, \qquad O \sim f(\cdot \mid I), \qquad L = \phi_t(I, O),$$

where the notation $O \sim f(\cdot \mid I)$ allows randomized model outputs. When $\mathcal{L}_t \subseteq \mathbb{R}$, we write the *task risk* as

$$\mu_t(f) := \mathbb{E}_{L \sim P_t^f}[L]. \tag{1}$$

*Example* 1 (Multi-environment evaluation). Many benchmarks aim to certify performance across multiple environments. Let $\mathcal{T}$ index environments. For each $t \in \mathcal{T}$, the item distribution $Q_t$ is over the input-label pairs $(X, Y)$. The model outputs $f(X)$, and the judge returns $L = \ell(f(X), Y)$ for some loss $\ell$. The distribution over $L$ induced by $Q_t$ and $f$ is the task-response distribution $P_t^f$.

*Example* 2 (Arithmetic evaluation). For arithmetic addition, an item is a pair $I = (a, b) \in \mathbb{N}^2$ with a prompt asking for $a + b$. The model outputs a string $O_f(I)$, and the judge returns the binary error $L = \mathbf{1}\{O_f(I) \neq a + b\}$. Each arithmetic task is indexed by $(a, b) \in \mathbb{N}^2$ and has a singleton item distribution $Q_{(a,b)} = \delta_{(a,b)}$. Thus the task-response distribution $P_{(a,b)}^f$ is the Bernoulli distribution induced by whether $f$ correctly computes $a + b$.

### 2.3. Capabilities

Let $\mathcal{T}$ be a (possibly infinite) *family of tasks* that may be encountered in the deployed environment. A *capability* with respect to $\mathcal{T}$ is defined as follows:

**Definition 2.1** (Capability). Given a task family $\mathcal{T}$, a capability is any functional $\mathcal{C}_{\mathcal{T}} : \mathcal{F} \to \mathcal{V}_{\mathcal{T}}$, where $\mathcal{C}_{\mathcal{T}}$ depends only on the collection of task-response laws $\{P_t^f : t \in \mathcal{T}\}$ induced by $f$ on the target tasks.

Intuitively, a model's capability with respect to a target task family $\mathcal{T}$ is a compact "summary" of how the model behaves when deployed across $\mathcal{T}$.

*Example* 3. Common capabilities include

- **Risk profile**: $\mathcal{C}_{\mathcal{T}}(f) = \{\mu_t(f)\}_{t \in \mathcal{T}}$, and

- **Worst-case risk**: $\mathcal{C}_{\mathcal{T}}(f) = \sup_{t \in \mathcal{T}} \mu_t(f)$,

where $\mu_t(f)$ is the task risk defined in (1).

*Remark* 2.2. A capability is an intrinsic property of a model with respect to an intended task family. It serves only as the object to be evaluated; by itself, it does not specify an evaluation strategy or protocol.

### 2.4. Evaluation protocols and generalization

An *evaluation protocol* is a procedure that queries a blackbox model $f$ on finitely many items, possibly outside the support of the target tasks, and aggregates the observable responses to *estimate* the target capability $\mathcal{C}_{\mathcal{T}}(f)$.

**Definition 2.3** (Evaluation protocol). An evaluation protocol $\Pi$ consists of (i) a possibly adaptive and randomized rule that selects query items $I_1, \ldots, I_n \in \mathcal{I}$, and (ii) an estimator that maps the transcript $(I_1, L_1), \ldots, (I_n, L_n)$ to an estimate $\widehat{\mathcal{C}} \in \mathcal{V}_{\mathcal{T}}$. The queried items may be sampled from the target tasks in $\mathcal{T}$, or chosen from outside the supports of the target task distributions.

**Definition 2.4** (Capability evaluation). Let $\varepsilon > 0$ and $\delta \in (0, 1)$. An evaluation protocol $\Pi$ $(\varepsilon, \delta)$-*evaluates* $\mathcal{C}_{\mathcal{T}}$ over $\mathcal{F}$ if for every $f \in \mathcal{F}$,

$$\Pr\left(d\left(\widehat{\mathcal{C}}, \mathcal{C}_{\mathcal{T}}(f)\right) \leq \varepsilon\right) \geq 1 - \delta,$$

where the probability is over all randomness and the randomness involved, and $d$ is a metric on $\mathcal{V}_{\mathcal{T}}$.

An evaluation protocol is $(\varepsilon, \delta)$-*generalizable* if, using a finite number of interactions with the model, it $(\varepsilon, \delta)$-evaluates the target capability. We refer to the minimum number of interactions sufficient to achieve this guarantee as the *sample (or evaluation) complexity*.

*Remark* 2.5. In this paper, we focus only on the intrinsic information-theoretic limits of capability evaluation. We assume the evaluator has full knowledge of the target task family $\mathcal{T}$ when designing $\Pi$. In practice, the evaluator's access model may be more limited, for example when the task family is only partially specified or must itself be learned. We leave such limited-access regimes for future work.

# 3. Main Results

In this section, we study several concrete *capability estimation* settings under the general framework introduced in Section 2. These settings are not meant to be exhaustive. Rather, they are chosen to demonstrate how careful formulation and structural assumptions enable precise theoretical characterizations of what can and cannot be inferred about a model's capabilities.

## 3.1. Stochastic Multi-Environment Evaluation

Perhaps the simplest and most widely used form of capability estimation, often adopted implicitly, is to evaluate a model's expected performance under a fixed data distribution using i.i.d. samples. In this regime, evaluation amounts to inferring the model's behavior on future draws from the *same* distribution. This paradigm underlies much of standard practice, including cross-validation, hold-out testing, and empirical risk estimation on benchmark datasets, and enjoys a crucial but often overlooked property: its validity relies solely on i.i.d. sampling and does not depend on any structural assumptions about the model under evaluation.

However, this single-distribution viewpoint tightly couples evaluation to the particular data-generating process used for testing. When models are deployed across multiple environments, heterogeneous populations, or shifting data sources, performance under one distribution provides no principled guarantee about behavior under others. This limitation motivates the study of *multi-distribution capability estimation*, in which evaluation must infer and certify a model's behavior across a collection (possibly infinite) of distinct distributions rather than a single one.

Formally, we consider a collection of data-generating processes (i.e., *environments*) $\mathcal{P} \subset \Delta(\mathcal{X} \times \mathcal{Y})$, where $\mathcal{X}$ is the input space and $\mathcal{Y}$ is the output space. Each $p \in \mathcal{P}$ represents a ground-truth data distribution. A model is an unknown stochastic predictor $f : \mathcal{X} \rightarrow \Delta(\mathcal{Y})$ (e.g., a LLM), and we assume a known loss function $\ell : \mathcal{Y} \times \mathcal{Y} \rightarrow [0, 1]$. For a data point $z = (x, y)$, we define the induced (random) loss $\ell_f(z) := \ell(f(x), y)$ and its conditional mean $\bar{\ell}_f(z) := \mathbb{E}[\ell_f(z) \mid z]$ [1]. The evaluation protocol interacts with the model only through the induced loss $\ell_f$.

An evaluation protocol, with knowledge of the environment set $\mathcal{P}$, generates a sequence of test data points $z_1, \ldots, z_n \in \mathcal{X} \times \mathcal{Y}$, possibly adaptively and using its own internal randomness. Upon querying these points, the protocol observes the corresponding losses $\ell_f(z_1), \ldots, \ell_f(z_n)$. The goal is to estimate the model's *risk profile*

$$\left\{ \mu_p(f) := \mathbb{E}_{z \sim p}[\bar{\ell}_f(z)] : p \in \mathcal{P} \right\},$$

that is, to infer the *expected* performance of $f$ under every environment in $\mathcal{P}$ simultaneously using only a finite collection of test samples.

An evaluation protocol $\mathcal{E}$ is said to be $(\varepsilon, \delta)$-*generalizable with sample size* $n$ if the following holds. For every model $f : \mathcal{X} \rightarrow \Delta(\mathcal{Y})$ and loss $\ell : \mathcal{Y} \times \mathcal{Y} \rightarrow [0, 1]$,

$$\Pr\left(\sup_{p \in \mathcal{P}}\left|\widehat{\ell}_f(p) - \mathbb{E}_{z \sim p}[\bar{\ell}_f(z)]\right| \geq \varepsilon\right) \leq \delta,$$

where the probability is taken over the randomness of the queried data points $z_1, \ldots, z_n$ generated by $\mathcal{E}$ and $\widehat{\ell}_f(p)$ denotes the estimate of $\mathbb{E}_{z \sim p}[\bar{\ell}_f(z)]$ produced by $\mathcal{E}$ using the observed transcript $\left\{(z_1, \ell_f(z_1)), \ldots, (z_n, \ell_f(z_n))\right\}$.

### 3.1.1. EVALUATION PROTOCOLS FOR FINITE $\mathcal{P}$

We first consider the case where the environment class $\mathcal{P}$ is finite, with $|\mathcal{P}| = m$. The following proposition shows that, without additional structure relating the environments, capability estimation cannot substantially outperform evaluating each environment independently.

**Proposition 3.1** (Lower bound for finite $\mathcal{P}$). *There exists a finite environment class $\mathcal{P} = \{p_1, \ldots, p_m\}$ such that any $(\varepsilon, \delta)$-generalizable evaluation protocol must use $n = \Omega(m/\varepsilon^2)$ samples, for any fixed $\delta < 1/2$.*

*Proof.* Let $\mathcal{X} = \{x_1, \ldots, x_m\}$ and $\mathcal{Y} = \{0, 1\}$, and use the 0–1 loss $\ell(y', y) = \mathbf{1}\{y' \neq y\}$. For each $i \in [m]$, define environment $p_i$ by assigning $(X, Y) = (x_i, 0)$ with probability 1. For a *stochastic* model $f$, write $f(x) \in \Delta(\{0, 1\})$ for its output distribution, and let $\ell_f(x)$ denote the induced (random) loss under $Y \equiv 0$. Then

$$\mu_i(f) := \mathbb{E}_{z \sim p_i}[\bar{\ell}_f(z)] = \Pr_{O \sim f(x_i)}[O = 1].$$

---

[1] Expectation over the model's internal randomness.

Fix $\varepsilon \in (0, 1/8)$. We draw a random stochastic model $f$ as follows: independently for each $i \in [m]$, sample $S_i \in \{+1, -1\}$ uniformly and set [2]

$$f(x_i) = \text{Bern}\left(\tfrac{1}{2} + 2\varepsilon S_i\right)$$

Under this construction, $\mu_i(f) = \tfrac{1}{2} + 2\varepsilon S_i$, and the $\{\mu_i(f)\}_{i=1}^m$ are independent. In particular, any estimator satisfying $|\widehat{\mu}_i - \mu_i(f)| \le \varepsilon$ must identify the sign $S_i$.

Let $\mathcal{E}$ be any (possibly adaptive and randomized) protocol that makes $n$ total queries to $f$, and let $N_i$ be the (random) number of queries issued at $x_i$. Then $\sum_{i=1}^m N_i = n$ almost surely, so $\sum_{i=1}^m \mathbb{E}[N_i] = n$ and there exists $i^\star$ with $\mathbb{E}[N_{i^\star}] \le n/m$. Conditioned on $S_{i^\star}$, the observations from queries at $x_{i^\star}$ are i.i.d. Bernoulli with mean $\tfrac{1}{2} \pm 2\varepsilon$, and $N_{i^\star}$ is a (data-dependent) stopping time. By the sequential testing lower bound (Corollary A.2), identifying $S_{i^\star}$ with constant success probability requires $\mathbb{E}[N_{i^\star}] = \Omega(1/\varepsilon^2)$. Therefore, if $n \ll m/\varepsilon^2$, then $\mathbb{E}[N_{i^\star}] \ll 1/\varepsilon^2$, and $\mathcal{E}$ fails to estimate $\mu_{i^\star}(f)$ within error $\varepsilon$ with constant probability under the above random draw of $f$. $\square$

*Remark* 3.2. A simple protocol achieves sample complexity $O\left(\frac{m \log(m/\delta)}{\varepsilon^2}\right)$ by drawing $O(\log(m/\delta)/\varepsilon^2)$ i.i.d. samples from each environment $p_i$, producing the empirical risk estimate for each $\mu_i(f)$, and applying a union bound to guarantee $\sup_{i \in [m]} |\widehat{\mu}_i - \mu_i(f)| \le \varepsilon$ w.p. $\ge 1 - \delta$.

*Remark* 3.3. The lower bound in Proposition 3.1 continues to hold even when the goal is only to estimate the *worst-case* risk $\max_{i \in [m]} \mu_i(f)$ up to error $\varepsilon$. In our construction, the maximizing environment is unknown and may be any block, so identifying the worst case necessarily requires resolving the same hard Bernoulli mean estimation instances. Thus, without additional structure on $\mathcal{P}$, estimating the worst-case risk is no easier than estimating the full risk profile.

Now, a natural question is what *structural* property of the family $\mathcal{P}$ allows for capability estimation beyond the pessimistic $\Omega(m/\varepsilon^2)$ lower bound. We show that when the environments admit a common proposal distribution with bounded $\chi^2$ divergence, the sample complexity depends on the resulting $\chi^2$-radius rather than on $m$. To formalize this notion, consider a family $\mathcal{P} = \{p_1, \ldots, p_m\}$ of distributions on a common domain $\mathcal{Z}$, and define the $\chi^2$-*radius* [3]

$$V(\mathcal{P}) := \inf_{q \in \Delta(\mathcal{Z}):\, p_i \ll q \,\forall i} \max_{i \in [m]} \chi^2(p_i \| q), \qquad (2)$$

where the $\chi^2$ divergence is given by

$$\chi^2(p \| q) := \int \left(\frac{dp}{dq} - 1\right)^2 dq. \qquad (3)$$

The following theorem shows that $V(\mathcal{P})$ controls the sample complexity of multi-distribution evaluation.

---

[2] $\text{Bern}(r)$ denotes Bernoulli distribution with parameter $r$.

[3] Also known as Rényi information radius in the literature.

**Theorem 3.4** ($\chi^2$-radius upper bound). *Let $\mathcal{P} = \{p_1, \ldots, p_m\}$ be a collection of probability distributions on a common domain $\mathcal{Z}$ with $\chi^2$-radius $V(\mathcal{P})$. Then there exists an evaluation protocol that is $(\varepsilon, \delta)$-generalizable for the full risk profile $\{\mu_i(f)\}_{i=1}^m$ with sample size*

$$n = O\left(\frac{(1 + V(\mathcal{P})) \log(m/\delta)}{\varepsilon^2}\right).$$

*Proof.* Fix a distribution $q$ attaining (or approximating) the infimum in the definition of $V(\mathcal{P})$. The evaluation protocol draws $Z_1, \ldots, Z_n \overset{\text{i.i.d.}}{\sim} q$ and queries the model to obtain $\ell_f(Z_j) \in [0, 1]$ for each $j \in [n]$.

For each environment $p_i$, define the importance-weighted random variables

$$Y_t^{(i)} := \frac{p_i(Z_t)}{q(Z_t)}\, \ell_f(Z_t).$$

Since $p_i \ll q$, these variables are well defined. Moreover,

$$
\begin{aligned}
\mathbb{E}[Y_t^{(i)}] &= \mathbb{E}_{z \sim q}\left[\frac{p_i(z)}{q(z)} \mathbb{E}[\ell_f(z) \mid z]\right] \\
&= \mathbb{E}_{z \sim p_i}[\bar{\ell}_f(z)] =: \mu_i(f).
\end{aligned}
$$

We next bound their second moments. Since $\ell_f(z) \in [0, 1]$ almost surely,

$$
\begin{aligned}
\mathbb{E}[(Y_t^{(i)})^2] &\le \mathbb{E}_{z \sim q}\left[\left(\frac{p_i(z)}{q(z)}\right)^2\right] \\
&= 1 + \chi^2(p_i \| q) \le 1 + V(\mathcal{P}).
\end{aligned}
$$

Hence $\text{Var}(Y_t^{(i)}) \le 1 + V(\mathcal{P})$ for all $i$.

To obtain uniform high-probability guarantees over $i \in [m]$, we use a median-of-means estimator. Partition the samples into $B = \lceil c \log(2m/\delta) \rceil$ disjoint blocks of equal size $s = \lfloor n/B \rfloor$. For each $i$, compute the block averages

$$\bar{Y}_b^{(i)} := \frac{1}{s} \sum_{t \in \text{block } b} Y_t^{(i)}, \qquad b = 1, \ldots, B,$$

and output $\widehat{\mu}_i$ as the median of $\{\bar{Y}_b^{(i)}\}_{b=1}^B$.

By Chebyshev's inequality, for any fixed $i$ and block $b$,

$$\Pr\left(|\bar{Y}_b^{(i)} - \mu_i(f)| > \varepsilon\right) \le \frac{1 + V(\mathcal{P})}{s \varepsilon^2}.$$

Choosing $s \ge 8(1 + V(\mathcal{P}))/\varepsilon^2$ ensures this probability is at most $1/8$. A standard median-of-means argument then yields

$$\Pr(|\widehat{\mu}_i - \mu_i(f)| > \varepsilon) \le \delta/m.$$

Finally, applying a union bound over $i \in [m]$ gives

$$\Pr\left(\max_{i \in [m]} |\widehat{\mu}_i - \mu_i(f)| > \varepsilon\right) \le \delta.$$

The stated sample complexity follows from $n = Bs$. $\square$

*Example* 4 (Simple bounds on the $\chi^2$-radius). Let $\mathcal{P} = \{p_1, \ldots, p_m\} \subset \Delta(\mathcal{Z})$. The following choices of $q$ give immediate upper bounds on $V(\mathcal{P}) := \inf_q \max_i \chi^2(p_i \| q)$.

(1) Taking $q = \frac{1}{m} \sum_{i=1}^m p_i$ yields $V(\mathcal{P}) \le m$.

(2) If $\mathcal{Z}$ is finite, taking $q = \text{Unif}(\mathcal{Z})$ yields $V(\mathcal{P}) \le |\mathcal{Z}|$.

(3) Let $S(\mathcal{P}) := \sum_{z \in \mathcal{Z}} \sup_i p_i(z)$ be the *Shtarkov sum* (Drmota & Szpankowski, 2023; Wu et al., 2026), and let $q(z) = \frac{\sup_i p_i(z)}{S(\mathcal{P})}$ be the normalized maximum-likelihood (NML) distribution (Cesa-Bianchi & Lugosi, 2006). Then $V(\mathcal{P}) \le S(\mathcal{P}) - 1$.

We complement the upper bound in Theorem 3.4 with the following lower bound, showing that the dependence on the $\chi^2$-radius is unavoidable for a broad class of natural evaluation protocols.

**Theorem 3.5** ($\chi^2$-radius lower bound). *Let* $\mathcal{P} = \{p_1, \ldots, p_m\}$ *be distributions on a common domain* $\mathcal{Z}$, *and fix a proposal distribution* $q$ *such that* $p_i \ll q$ *for all* $i$. *Define* $S_q(\mathcal{P}) := \max_{i \in [m]} \mathbb{E}_q[(p_i/q)^2]$. *Consider any evaluation protocol that draws samples i.i.d. from* $q$ *and observes the corresponding losses. Suppose*

$$\max_{i \in [m]} \operatorname*{ess\,sup}_{z \in \mathcal{Z}} \frac{p_i(z)}{q(z)} \le \frac{c\, S_q(\mathcal{P})}{\varepsilon}$$

*for a universal constant* $c > 0$. *Then for any fixed* $\delta < 3/8$, *achieving* $(\varepsilon, \delta)$-*generalizability requires*

$$n = \Omega\left(\frac{S_q(\mathcal{P})}{\varepsilon^2}\right).$$

*In particular, since* $S_q(\mathcal{P}) \ge 1 + V(\mathcal{P})$ *for every* $q$, *this implies the lower bound* $n = \Omega(V(\mathcal{P})/\varepsilon^2)$.

*Proof sketch.* Fix the proposal distribution $q$ used by the protocol, and choose an index $i^\star$ such that $S_q(\mathcal{P}) = \max_i \mathbb{E}_q[(p_i/q)^2] = 1 + \chi^2(p_{i^\star} \| q)$. We construct two stochastic models $f^+$ and $f^-$ whose induced losses differ only through their behavior on samples drawn from $q$. Specifically, given a query point $Z \sim q$, the observed loss is Bernoulli with parameter $\frac{1}{2} \pm \theta(Z)$, where $\theta(Z)$ is chosen proportional to the likelihood ratio $p_{i^\star}(Z)/q(Z)$ and scaled so that $|\theta(Z)| \le 1/4$ using the assumed likelihood-ratio bound. By construction, the conditional mean losses satisfy $\bar{\ell}_{f^+}(Z) - \bar{\ell}_{f^-}(Z) = 2\theta(Z)$, which implies that the induced risks under environment $p_{i^\star}$ satisfy $\mu_{i^\star}(f^+) - \mu_{i^\star}(f^-) = \mathbb{E}_{z \sim p_{i^\star}}[\bar{\ell}_{f^+}(z) - \bar{\ell}_{f^-}(z)] = \Theta(\varepsilon)$.

On the other hand, since the protocol observes i.i.d. losses generated from $Z_t \sim q$, the KL divergence between the induced transcript distributions satisfies $\text{KL}(\mathbb{P}_{f^+} \| \mathbb{P}_{f^-}) = O(n \mathbb{E}_{z \sim q}[\theta(z)^2]) = O(n \varepsilon^2/S_q(\mathcal{P}))$, where the last step

uses the chosen scaling of $\theta$ and the definition of $S_q(\mathcal{P})$. Therefore, unless $n = \Omega(S_q(\mathcal{P})/\varepsilon^2)$, the two models are statistically indistinguishable (by Pinsker's and Le Cam's inequality), implying a constant probability of error for any estimator of $\mu_{i^\star}(f)$ at accuracy $\varepsilon$. The full proof formalizes this two-point argument and yields the stated $\Omega(S_q(\mathcal{P})/\varepsilon^2)$ lower bound, which in particular implies $\Omega(V(\mathcal{P})/\varepsilon^2)$ since $S_q(\mathcal{P}) \ge 1 + V(\mathcal{P})$. See Appendix B for full proof. $\square$

Note that the likelihood-ratio condition in Theorem 3.5 holds for many natural proposal distributions, including the "balanced" proposals such as those in Example 4 (for sufficiently small $\varepsilon$). Moreover, in Appendix D we show that the same $\Omega(V(\mathcal{P})/\varepsilon^2)$ lower bound extends to *fully adaptive evaluation protocols*, provided that the query distribution they induce (measured under an appropriate null reference model) satisfies an analogous overlap condition with the environments. In both settings, this likelihood-ratio condition informally requires that no environment in $\mathcal{P}$ places substantially more probability mass than the protocol's effective proposal on vanishingly rare regions of the domain.

At the same time, some form of tail control appears necessary if one wants lower bounds purely in terms of $\chi^2(p_i \| q)$. Let $q$ be the proposal distribution (or, in the adaptive case, the reference distribution over queries induced by the protocol). If an environment $p_i$ differs from $q$ only on inputs that are extremely unlikely under $q$, then $p_i(z)/q(z)$ can be huge on that tiny set, causing $\chi^2(p_i \| q)$ to blow up even when the induced risk is negligible. Concretely, fix $\varepsilon > 0$, choose $\eta \in (0, \varepsilon/10]$, set $\delta = \eta^3$, and let $p_i$ agree with $q$ everywhere except on an environment-specific region $B_i$ with $p_i(B_i) = \eta$ and $q(B_i) = \delta$; since $\bar{\ell}_f$ is bounded, the contribution of $B_i$ to $\mathbb{E}_{z \sim p_i}[\bar{\ell}_f(z)]$ is at most $\eta \le \varepsilon/10$. Yet on $B_i$ we have $p_i/q = \eta/\delta = 1/\eta^2$, so $\chi^2(p_i \| q) = \mathbb{E}_{z \sim q}[(p_i/q - 1)^2] \ge q(B_i)(\eta/\delta - 1)^2 = \delta(1/\eta^2 - 1)^2 \ge 1/(4\eta)$ for $\eta \le 1/\sqrt{2}$, which can be arbitrarily large as $\eta \to 0$. We expect that settling whether the $\chi^2$-radius lower bound holds *unconditionally* will require fundamentally new techniques.

*Remark* 3.6. Importance-weighting ideas related to Theorem 3.4 appear in the multi-distribution mean estimation literature (e.g., (Elvira et al., 2015; Demange-Chryst et al., 2023)), which largely emphasizes variance reduction. In contrast, we focus on *uniform, high-probability* guarantees across environments and characterize the resulting sample complexity. To the best of our knowledge, our formulation of the $\chi^2$-radius as the governing complexity parameter for multi-environment *capability evaluation*, together with matching-order lower-bound analysis (under a natural overlap condition), has not been previously identified.

### 3.1.2. EVALUATION PROTOCOLS FOR INFINITE $\mathcal{P}$

The upper bound in Theorem 3.4 scales as $O((1 + V(\mathcal{P}))\log(m/\delta)/\varepsilon^2)$ when $|\mathcal{P}| = m$. At first glance, the $\log m$ factor may appear to be a proof artifact from a union bound. The next proposition shows that this dependence is in fact unavoidable, even when the $\chi^2$-radius is a *constant*.

**Proposition 3.7** (A necessary $\log|\mathcal{P}|$ dependence). *For every integer $m \geq 2$ there exist a finite domain $\mathcal{Z}$ and a family $\mathcal{P} \subset \Delta(\mathcal{Z})$ with $V(\mathcal{P}) \leq 1$ and $|\mathcal{P}| = m$ such that, for any fixed $\delta \in (0, 1/6)$, every (possibly adaptive and randomized) evaluation protocol that is $(\varepsilon, \delta)$-generalizable for the full risk profile must use $\Omega\left(\frac{\log m}{\varepsilon^2}\right)$ queries.*

*Proof sketch.* Fix an integer $m \geq 2$ and set $D := \lceil c \log m \rceil$ for a sufficiently large absolute constant $c$. Let $\mathcal{Z} = [D]$ and let $q$ be uniform on $\mathcal{Z}$. For each half-set $A \subseteq [D]$ with $|A| = D/2$, define $p_A(z) = \frac{2}{D}\mathbf{1}\{z \in A\}$. Then $p_A/q \in \{0, 2\}$ and $\chi^2(p_A\|q) = 1$, hence $V(\mathcal{P}) = O(1)$.

Choose $m$ half-sets $A_1, \ldots, A_m$ with pairwise overlaps close to $D/4$. For each $j \in [m]$, define a *randomized* model $f^{(j)}$ that on query $z$ returns a Bernoulli loss with mean $\frac{1}{2} + \frac{\varepsilon}{2}(2\mathbf{1}\{z \in A_j\} - 1)$. Then $\mu_{A_j}(f^{(j)}) = \frac{1}{2} + \varepsilon/2$, while for $k \neq j$ one has $\mu_{A_k}(f^{(j)}) \in [\frac{1}{2} - \varepsilon/8, \frac{1}{2} + \varepsilon/8]$. Thus estimating the full risk profile to accuracy $O(\varepsilon)$ identifies the hidden index $j$.

For any adaptive query strategy, each observation provides only $O(\varepsilon^2)$ information about the hidden index $j$ (a $\Theta(\varepsilon)$-biased coin flip). Hence $n$ queries yield at most $O(n\varepsilon^2)$ information. By Fano's inequality, identifying one of $m$ hypotheses with constant error requires $\Omega(\log m)$ information, so $n = \Omega((\log m)/\varepsilon^2)$. Details are in Appendix C. $\qquad\square$

The preceding proposition shows that a $\log|\mathcal{P}|$ dependence is unavoidable for *large* finite classes. For genuinely infinite classes, the correct analogue is to replace $|\mathcal{P}|$ by an appropriate covering number. The next theorem gives an upper bound obtained by combining (i) a total-variation cover of $\mathcal{P}$ and (ii) the finite-class guarantee on the cover elements.

**Theorem 3.8** (Upper bound for infinite classes via total–variation covering). *Let $\mathcal{P}$ be a class of distributions on a common domain $\mathcal{Z}$. Fix $\varepsilon \in (0, 1)$ and assume that $\mathcal{P}$ admits a finite cover $\mathcal{N} \subset \Delta(\mathcal{Z})$ in total variation with radius $\varepsilon/2$, i.e., for every $p \in \mathcal{P}$ there exists $\nu(p) \in \mathcal{N}$ such that*

$$\mathrm{TV}(p, \nu(p)) \leq \varepsilon/2.$$

*Then there exists an evaluation protocol that is $(\varepsilon, \delta)$-generalizable for the full risk profile $\{\mathbb{E}_{z\sim p}[\bar{\ell}_f(z)] : p \in \mathcal{P}\}$ with sample complexity*

$$O\left(\frac{(1 + V(\mathcal{N}))\log(|\mathcal{N}|/\delta)}{\varepsilon^2}\right),$$

*where $V(\mathcal{N})$ is the $\chi^2$-radius of the finite set $\mathcal{N}$.*

*Proof sketch.* Run the finite-class protocol of Theorem 3.4 on the net $\mathcal{N}$ to obtain estimates $\{\widehat{\mu}_\nu\}_{\nu\in\mathcal{N}}$ satisfying $\max_{\nu\in\mathcal{N}}|\widehat{\mu}_\nu - \mathbb{E}_{z\sim\nu}[\bar{\ell}_f(z)]| \leq \varepsilon/2$ with probability at least $1 - \delta$. For an arbitrary $p \in \mathcal{P}$, define $\widehat{\mu}_p := \widehat{\mu}_{\nu(p)}$. Since $\ell_f \in [0, 1]$, total variation controls expectation differences:

$$\left|\mathbb{E}_p[\bar{\ell}_f] - \mathbb{E}_{\nu(p)}[\bar{\ell}_f]\right| \leq \mathrm{TV}(p, \nu(p)) \leq \varepsilon/2.$$

Combining the two inequalities yields $|\widehat{\mu}_p - \mathbb{E}_p[\bar{\ell}_f]| \leq \varepsilon$ uniformly over $p \in \mathcal{P}$. $\qquad\square$

We now apply Theorem 3.8 to illustrate how one can estimate risks for *uncountably many* distributions using a finite evaluation budget. We consider the length-$k$ Bernoulli product family on $\{0, 1\}^k$, for which the resulting bound depends nontrivially on $k$. The key step is that the $\chi^2$-radius of a suitable total-variation cover can be upper bounded by the Shtarkov sum of the Bernoulli source.

**Corollary 3.9** (Bernoulli i.i.d. sources of length $k$). *Let $\mathcal{P} = \{\mathrm{Bern}(\theta)^{\otimes k} : \theta \in [0, 1]\}$ be the class of length-$k$ i.i.d. Bernoulli distributions on $\mathcal{Z} = \{0, 1\}^k$. Assume only that the induced loss $\ell_f : \mathcal{Z} \to \Delta([0, 1])$ is arbitrary. Then there exists an evaluation protocol that is $(\varepsilon, \delta)$-generalizable for $\mathcal{P}$ with sample complexity*

$$O\left(\frac{\sqrt{k}\,\log\left(\frac{k}{\varepsilon\delta}\right)}{\varepsilon^2}\right).$$

*Proof sketch.* Let $p_\theta = \mathrm{Bern}(\theta)^{\otimes k}$. A standard coupling bound gives $\mathrm{TV}(p_\theta, p_{\theta'}) \leq k|\theta - \theta'|$. Hence the grid $\Theta = \{0, \Delta, 2\Delta, \ldots, 1\}$ with $\Delta = \varepsilon/(2k)$ yields a TV-cover $\mathcal{N} := \{p_\theta : \theta \in \Theta\}$ of radius $\varepsilon/2$ and size $|\mathcal{N}| = O(k/\varepsilon)$.

By Example 4(3), $V(\mathcal{N}) \leq S(\mathcal{N})$ where $S(\mathcal{N})$ is the Shtarkov sum. Since $\mathcal{N} \subseteq \{p_\theta : \theta \in [0, 1]\}$, we have $S(\mathcal{N}) \leq S_k$, where

$$S_k := \sum_{z \in \{0,1\}^k} \sup_{\theta \in [0,1]} p_\theta(z)$$

is the Shtarkov sum of the length-$k$ Bernoulli source. It is known that $S_k = \Theta(\sqrt{k})$ (see, e.g., (Szpankowski, 1998; Cesa-Bianchi & Lugosi, 2006)). Therefore $V(\mathcal{N}) = O(\sqrt{k})$, and Theorem 3.8 with $|\mathcal{N}| = O(k/\varepsilon)$ gives the claimed sample complexity. $\qquad\square$

Theorem 3.8 highlights a useful two-stage approach. A TV-cover controls approximation error when replacing an infinite class by a finite proxy, but by itself *does not* capture how efficiently the proxy can be evaluated (the cover may have exponentially many elements). Conversely, the $\chi^2$-radius controls the evaluation complexity of a finite family, but

by itself *does not* address infinite classes (Proposition 3.7). Taken together, these two ingredients yield sharp and sometimes genuinely nontrivial guarantees, as the Bernoulli example illustrates: the bound is driven neither by covering alone nor by $\chi^2$-radius alone, but by their interaction.

## 3.2. Combinatorial Evaluation with Structured Models

In Section 3.1, we analyzed capability evaluation in stochastic environments. In this regime, the objective is inherently statistical, i.e., to estimate *expected* performance under each environment. With i.i.d. samples (from each environment or a proposal distribution), one can obtain PAC-style guarantees *without* assumptions on the model class.

We now shift to a different and equally important paradigm in which the evaluation goal is *combinatorial* rather than stochastic. Here the target is not an expected risk, but a stronger form of certification, such as a worst-case guarantee, exact correctness on an entire task family, or a promise that the model makes at most a bounded number of errors. This regime arises naturally when a capability is intended to be "algorithmic" or "rule-based"—for instance, when one wants to certify whether an LLM can reliably perform arithmetic, obey a formal specification, or follow a syntactic constraint across all relevant inputs. In such settings, evaluation cannot rely on averaging under a fixed distribution: failures may be rare yet consequential, and a small number of adversarially placed errors can invalidate a capability claim.

A fundamental difference from the stochastic regime is that meaningful certification is impossible without structural restrictions on the *model class*. Indeed, for a completely unrestricted model, any finite evaluation transcript can be matched by another model that behaves identically on the queried tasks but differs arbitrarily on unqueried ones. We illustrate the phenomenon on Boolean circuits with bounded size. Let $\mathcal{F}_{n,s}$ be the class of Boolean functions $f : \{0,1\}^n \to \{0,1\}$ computable by circuits of size at most $s$. The task family is $\mathcal{T} = \{(x,0) : x \in \{0,1\}^n\}$ with indicator loss. We consider the counting capability

$$\mathcal{C}(f) := \big|\{x \in \{0,1\}^n : f(x) = 1\}\big|.$$

An evaluator is given black-box query access to $f$ and outputs an estimate $\widehat{\mathcal{C}}$. If we interpret $f(x) = 1$ as indicating a failure on input $x$ (i.e., the model produces an incorrect answer on that task instance), then $\mathcal{C}(f)$ measures the total number of failing inputs. We show that, unlike the stochastic setting where guarantees are largely *model-class-agnostic*, this combinatorial objective admits no such generalization: even for $f \in \mathcal{F}_{n,s}$, any black-box protocol using subexponentially many queries (more than the circuit size $s$), can be forced to incur additive error $M(s)$.

**Theorem 3.10.** *There exists a constant $c > 0$ such that for*

*all sufficiently large $n$ and all $s \geq cn$, the following holds. For any functions $T, M : \mathbb{N} \to \mathbb{N}$ satisfying $T(n)(2M(s) + 1) \leq 2^{n-2}$, no (possibly randomized, adaptive) evaluator making at most $T(n)$ queries can, with success probability at least $2/3$, output an estimate $\widehat{\mathcal{C}}$ satisfying*

$$\big|\widehat{\mathcal{C}} - \mathcal{C}(f)\big| \leq M(s) \quad \text{for all } f \in \mathcal{F}_{n,s}.$$

*Proof.* We apply Yao's minimax principle, so it suffices to exhibit a distribution over functions in $\mathcal{F}_{n,s}$ under which every deterministic evaluator with at most $T(n)$ queries fails with probability at least $1/3$.

Identify $\{0,1\}^n$ with the integers $[2^n] := \{0, 1, \ldots, 2^n - 1\}$ via the standard binary encoding. Let $W := 2M(s) + 1$. Consider the following random choice of $f$: with probability $1/2$ set $f = f_0 \equiv 0$; with probability $1/2$ choose a uniformly random interval $I = \{a, a+1, \ldots, a+W-1\} \subseteq [2^n]$ [4] (where $a$ is uniform in $\{0, \ldots, 2^n - W\}$) and set

$$f = f_I, \qquad f_I(x) = \mathbf{1}\{x \in I\}.$$

Then $\mathcal{C}(f_0) = 0$ and $\mathcal{C}(f_I) = |I| = W = 2M(s) + 1$.

Fix an arbitrary deterministic evaluator making at most $T(n)$ queries, and let $Q_0 \subseteq [2^n]$ be the (deterministic) set of queried inputs under $f_0$. The probability (over the random interval $I$) that $Q_0$ intersects $I$ is at most

$$\Pr(I \cap Q_0 \neq \emptyset) \ \leq \ \frac{|Q_0| \cdot |I|}{2^n} \ \leq \ \frac{T(n)\, W}{2^n}.$$

By the assumed condition $T(n)(2M(s) + 1) \leq 2^{n-2}$, we have $T(n)W/2^n \leq 1/4$. Hence, with probability at least $3/4$, the evaluator never queries a point in $I$; thus, the observed transcript is identically 0 under both $f_0$ and $f_I$.

Condition on the event that the transcript is all zeros, and let the evaluator output some value $a$. If $a \leq M(s)$ then $|\widehat{\mathcal{C}} - \mathcal{C}(f_I)| = |a - (2M(s)+1)| > M(s)$, while if $a > M(s)$ then $|\widehat{\mathcal{C}} - \mathcal{C}(f_0)| = |a - 0| > M(s)$. Thus, on the all-zero transcript event, the evaluator incurs error $> M(s)$ on at least one of $\{f_0, f_I\}$. Under the above $1/2$–$1/2$ mixture, this yields conditional failure probability at least $1/2$. Therefore the overall failure probability is at least $(3/4) \cdot (1/2) = 3/8 > 1/3$.

It remains to verify realizability. The function $f_I(x) = \mathbf{1}\{a \leq x \leq a + W - 1\}$ can be implemented as the AND of two comparators ("$x \geq a$" and "$x \leq a + W - 1$") with hardwired constants. Standard comparator circuits have size $O(n)$, so for $s \geq cn$ (with $c$ sufficiently large) we have $f_I \in \mathcal{F}_{n,s}$. By Yao's principle, the same lower bound holds for randomized evaluators. $\square$

---

[4]The intervals ensure $f_I$ admits a simple $O(n)$-size realization; an arbitrary set $I$ need not admit such a small circuit.

Theorem 3.10 identifies a fundamental information-theoretic limitation of black-box evaluation in the combinatorial regime. It shows that even when the evaluator is allowed to make *far more oracle queries than the circuit size $s$*, and indeed more than the model's description length, no black-box procedure can reliably estimate the capability measure $\mathcal{C}(f)$ to within additive error $M(s)$ for all $f \in \mathcal{F}_{n,s}$, where the tolerance $M(s)$ can scale with the circuit size, as long as the total number of queries satisfies $T(n) = o(2^n/M(s))$. This contrasts sharply with stochastic settings, where sample complexity is often model-class independent. We expect analogous black-box impossibility phenomena to extend to neural networks and transformer-based models as well.

## 4. Conclusion and Discussion

This paper frames evaluation as *certification by inference*. Our results show that meaningful generalization from finite evaluation is possible only when there is usable structure linking the task family and the model under evaluation. In the stochastic multi-environment setting, we show that when environments overlap, the sample complexity of certifying the full risk profile is governed by an intrinsic overlap parameter (the $\chi^2$ radius) rather than by the number of environments. In contrast, our combinatorial results show that without explicit structure linking tested and untested queries, black-box evaluation cannot reliably distinguish perfect performance from performance with many errors. Together, these results demonstrate a sharp boundary between what finite evaluation can and cannot certify.

There are several open problems left unresolved. For instance, our $\chi^2$-radius lower bound relies on a likelihood-ratio condition imposed on the evaluation protocol itself. It would be interesting to investigate whether this condition can be removed, and more generally whether an alternative complexity measure, such as a "truncated" version of the $\chi^2$-radius, could close the remaining gap between upper and lower bounds. For the combinatorial setting, our results are primarily negative. That said, these results do not rule out the existence of nontrivial positive guarantees for specific structured model classes. For example, consider the class of parity functions

$$\mathcal{F} := \{\, f(x) = \bigoplus_{i \in S} x_i \;:\; S \subseteq [n] \,\}.$$

For this family, simple random sampling suffices to distinguish between perfect performance (e.g., $\mathcal{C}(f) = 0$) and nontrivial error, since any incorrect parity function disagrees with the true function on exactly half of the inputs and can therefore be detected using i.i.d. samples from the uniform distribution on $\{0,1\}^n$. It is therefore an interesting direction to investigate finer-grained structural properties of the model class and the task family that enable combinatorial certifications.

Our findings also point toward directions for making practical evaluation claims more precise. In particular, our theory suggests that certification guarantees should clearly state the assumptions on the task family under which they are intended to hold, since such assumptions play a central role in enabling nontrivial inference. More broadly, it can be useful to view evaluation as a sequential decision problem, in which queries are chosen to probe the target capability, rather than relying solely on a single static test suite.

## Acknowledgments

This work is partially supported by the NSF Center for Science of Information (CSoI) Grant CCF-0939370, and also by NSF Grants CCF-2006440 and and CCF-2211423.

## Impact Statement

This paper presents work whose goal is to advance the field of machine learning. There are many potential societal consequences of our work, none of which we feel must be specifically highlighted here.

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

# A. A Sequential Lower Bound for Bernoulli Mean Testing (Optional Stopping)

This appendix states and proves a standard sequential (optional-stopping) lower bound used in Proposition 3.1. The key point is that the stopping time may depend on the observed samples, yet distinguishing two Bernoulli means still requires $\Omega(1/\varepsilon^2)$ *expected* samples. Closely related inequalities appear, for example, in (Kaufmann et al., 2016); we include a self-contained proof here for completeness and readability.

## A.1. Setup

Let $P$ and $Q$ be two distributions on a measurable space $(\mathcal{X}, \mathcal{A})$. Let $X_1, X_2, \ldots$ be i.i.d. samples drawn either from $P$ or from $Q$. Let $\mathcal{F}_t = \sigma(X_1, \ldots, X_t)$ be the natural filtration. A (possibly randomized) sequential test consists of: (i) a stopping time $\tau$ with respect to $(\mathcal{F}_t)_{t \geq 1}$, (ii) a terminal decision $\psi \in \{0, 1\}$ that is $\mathcal{F}_\tau$-measurable. We interpret $\psi = 0$ as deciding $P$ and $\psi = 1$ as deciding $Q$.

If the procedure uses internal randomness $U$ (independent of the samples), we enlarge the filtration to $\mathcal{G}_t = \sigma(U, X_1, \ldots, X_t)$ and require $\tau$ to be a stopping time w.r.t. $(\mathcal{G}_t)$ and $\psi$ to be $\mathcal{G}_\tau$-measurable. All statements below hold conditional on $U$, and hence hold unconditionally; for readability we present the proof without $U$.

## A.2. Main inequality

**Lemma A.1** (Sequential KL lower bound). *Let $(\tau, \psi)$ be any sequential test such that $\mathbb{E}_P[\tau] < \infty$ and*

$$P(\psi = 1) \leq \delta, \qquad Q(\psi = 0) \leq \delta,$$

*for some $\delta \in (0, 1/2)$. Then*

$$\mathbb{E}_P[\tau] \cdot \mathrm{KL}(P\|Q) \geq \mathrm{kl}(1 - \delta, \delta),$$

*where $\mathrm{KL}(\cdot\|\cdot)$ is the Kullback–Leibler divergence and $\mathrm{kl}(a, b) = a \log \frac{a}{b} + (1 - a) \log \frac{1-a}{1-b}$ is the binary relative entropy. By symmetry, also $\mathbb{E}_Q[\tau] \cdot \mathrm{KL}(Q\|P) \geq \mathrm{kl}(1 - \delta, \delta)$.*

*Proof.* Define the per-sample log-likelihood ratio

$$Y_t := \log \frac{dP}{dQ}(X_t),$$

interpreting $Y_t = -\infty$ on sets where $dP/dQ = 0$ (the statement is trivial otherwise). Let

$$L_t := \sum_{s=1}^{t} Y_s, \qquad t \geq 1,$$

and define the stopped transcript

$$\mathsf{T} := (\tau, X_1, \ldots, X_\tau),$$

which takes values in the disjoint union $\bigcup_{t \geq 1} \{t\} \times \mathcal{X}^t$.

**Step 1: Likelihood ratio of the stopped transcript.** For each $t \geq 1$ and each $(x_1, \ldots, x_t) \in \mathcal{X}^t$, write

$$A_t(x_1, \ldots, x_t) := \mathbf{1}\{\tau(x_1, \ldots, x_t) = t\}.$$

Since $\tau$ is a stopping time, $A_t$ is $\mathcal{X}^t$-measurable. Moreover, $A_t$ is the same deterministic function of the observed data under both $P$ and $Q$. Therefore,

$$P(\tau = t, X_1 \in dx_1, \ldots, X_t \in dx_t) = A_t(x_1, \ldots, x_t) \prod_{s=1}^{t} P(dx_s),$$

and similarly with $Q$ in place of $P$. Taking the Radon–Nikodym derivative on $\{t\} \times \mathcal{X}^t$ yields, pointwise on the event $\{\tau = t\}$,

$$\frac{dP_\mathsf{T}}{dQ_\mathsf{T}}(t, x_1, \ldots, x_t) = \prod_{s=1}^{t} \frac{dP}{dQ}(x_s).$$

Equivalently, the log-likelihood ratio satisfies

$$\log \frac{dP_{\mathsf{T}}}{dQ_{\mathsf{T}}}(\mathsf{T}) = L_\tau. \tag{4}$$

Importantly, this identity does *not* require $P(\tau = t) = Q(\tau = t)$; the stopping indicator cancels *pathwise* in the likelihood ratio because $\tau$ is a deterministic function of the observed data.

**Step 2: KL of the stopped transcript.**   Taking expectation of (4) under $P$ gives

$$\mathrm{KL}(P_{\mathsf{T}}\|Q_{\mathsf{T}}) = \mathbb{E}_P[L_\tau].$$

**Step 3: Wald's identity (optional stopping for sums).**   Under $P$, the $Y_t$ are i.i.d. with mean $\mathbb{E}_P[Y_1] = \mathrm{KL}(P\|Q)$. Since $\mathbb{E}_P[\tau] < \infty$ and $\mathbb{E}_P[|Y_1|] < \infty$ for Bernoulli $P, Q$ (and more generally when $\mathrm{KL}(P\|Q) < \infty$), Wald's identity applies:

$$\mathbb{E}_P[L_\tau] = \mathbb{E}_P\left[\sum_{s=1}^{\tau} Y_s\right] = \mathbb{E}_P[\tau] \cdot \mathbb{E}_P[Y_1] = \mathbb{E}_P[\tau] \cdot \mathrm{KL}(P\|Q).$$

Hence,

$$\mathrm{KL}(P_{\mathsf{T}}\|Q_{\mathsf{T}}) = \mathbb{E}_P[\tau] \cdot \mathrm{KL}(P\|Q).$$

**Step 4: Data processing to the terminal decision.**   Since $\psi$ is a measurable function of the transcript $\mathsf{T}$, the data processing inequality yields

$$\mathrm{KL}(P_{\mathsf{T}}\|Q_{\mathsf{T}}) \geq \mathrm{KL}(P_\psi\|Q_\psi),$$

where $P_\psi$ and $Q_\psi$ denote the induced Bernoulli laws of $\psi$ under $P$ and $Q$. Let $a := P(\psi = 0) \geq 1 - \delta$ and $b := Q(\psi = 0) \leq \delta$. Then

$$\mathrm{KL}(P_\psi\|Q_\psi) = \mathrm{kl}(a, b) \geq \mathrm{kl}(1 - \delta, \delta),$$

since $\mathrm{kl}(a, b)$ is increasing as $a$ moves away from $b$ and the minimum over the constraints $a \geq 1 - \delta$, $b \leq \delta$ is attained at $a = 1 - \delta$, $b = \delta$.

Combining the above displays gives

$$\mathbb{E}_P[\tau] \cdot \mathrm{KL}(P\|Q) = \mathrm{KL}(P_{\mathsf{T}}\|Q_{\mathsf{T}}) \geq \mathrm{kl}(1 - \delta, \delta),$$

which proves the claim. The bound under $Q$ is identical with $(P, Q)$ swapped. $\square$

## A.3. Specialization to Bernoulli means

**Corollary A.2** (Bernoulli separation implies $\Omega(1/\varepsilon^2)$ expected samples). *Fix $\varepsilon \in (0, 1/8)$ and let $P = \mathrm{Bern}(\frac{1}{2} + 2\varepsilon)$ and $Q = \mathrm{Bern}(\frac{1}{2} - 2\varepsilon)$. Any sequential test with error probabilities at most $\delta < 1/2$ satisfies*

$$\max\{\mathbb{E}_P[\tau], \mathbb{E}_Q[\tau]\} = \Omega(1/\varepsilon^2),$$

*where the hidden constant depends only on $\delta$.*

*Proof.* By Lemma A.1 it suffices to upper bound $\mathrm{KL}(P\|Q)$ by $C\varepsilon^2$. Let $P = \mathrm{Bern}(p)$ and $Q = \mathrm{Bern}(q)$ with $p = \frac{1}{2} + 2\varepsilon$ and $q = \frac{1}{2} - 2\varepsilon$. For Bernoulli distributions one has the standard inequality

$$\mathrm{KL}(\mathrm{Bern}(p) \,\|\, \mathrm{Bern}(q)) \;\leq\; \frac{(p - q)^2}{q(1 - q)} \qquad \text{for all } p, q \in (0, 1).$$

Since $q \in [1/4, 3/4]$ for $\varepsilon \leq 1/8$, we have $q(1 - q) \geq 3/16$, and therefore

$$\mathrm{KL}(P\|Q) \;\leq\; \frac{(4\varepsilon)^2}{3/16} = \frac{256}{3}\,\varepsilon^2.$$

Plugging this into Lemma A.1 yields $\mathbb{E}_P[\tau] \geq c(\delta)/\varepsilon^2$, and similarly $\mathbb{E}_Q[\tau] \geq c(\delta)/\varepsilon^2$. $\square$

# B. A $\chi^2$-Radius Lower Bound for Fixed-Proposal Evaluation

This appendix proves Theorem 3.5. The proof is based on a two-point (Le Cam) argument tailored to the fixed-proposal evaluation setting, in which the evaluator draws i.i.d. samples from a proposal distribution $q$ and observes only the induced losses. We construct two *stochastic* models whose risks under a carefully chosen environment differ by $2\varepsilon$, yet whose induced observation distributions under $q$ have small KL divergence unless $n = \Omega(V(\mathcal{P})/\varepsilon^2)$.

## B.1. Preliminaries

Fix a measurable space $(\mathcal{Z}, \mathcal{A})$ and a proposal distribution $q$ on $\mathcal{Z}$. Assume $p \ll q$ and define the likelihood ratio $w(z) := \frac{dp}{dq}(z)$. We will use Pinsker's inequality between total variation and KL (Polyanskiy & Wu, 2025, Theorem 7.10),

$$\mathrm{TV}(P, Q) \leq \sqrt{\tfrac{1}{2}\mathrm{KL}(P\|Q)}. \tag{5}$$

We also use the following elementary reduction from estimation to testing.

**Lemma B.1** (From estimation to testing). *Fix $i \in [m]$ and consider two models $f^+, f^-$ such that $\mu_i(f^+) - \mu_i(f^-) > 2\varepsilon$. If an evaluation protocol outputs an estimate $\widehat{\mu}_i$ satisfying*

$$\Pr_f\Big(|\widehat{\mu}_i - \mu_i(f)| \leq \varepsilon\Big) \geq 1 - \delta \qquad \text{for all } f \in \{f^+, f^-\},$$

*where $\Pr_f$ denotes probability over the protocol transcript when querying $f$, then there exists a test $\psi$ (measurable w.r.t. the transcript) taking values in $\{+, -\}$ such that*

$$\Pr_{f^+}(\psi = -) \leq \delta \qquad \text{and} \qquad \Pr_{f^-}(\psi = +) \leq \delta.$$

*Proof.* Define $\psi$ by thresholding $\widehat{\mu}_i$ at the midpoint $t := \frac{1}{2}(\mu_i(f^+) + \mu_i(f^-))$: output $+$ if $\widehat{\mu}_i \geq t$ and output $-$ otherwise. If $f = f^+$ and $|\widehat{\mu}_i - \mu_i(f^+)| \leq \varepsilon$, then $\widehat{\mu}_i \geq \mu_i(f^+) - \varepsilon = \frac{1}{2}(\mu_i(f^+) + \mu_i(f^-)) + \frac{1}{2}(\mu_i(f^+) - \mu_i(f^-) - 2\epsilon) > t$, so $\psi = +$. Similarly, if $f = f^-$ and $|\widehat{\mu}_i - \mu_i(f^-)| \leq \varepsilon$, then $\widehat{\mu}_i \leq \mu_i(f^-) + \varepsilon < t$, so $\psi = -$. Hence each error event is contained in the corresponding estimation failure event, implying the claimed bounds. $\square$

## B.2. Hard instance for a fixed proposal $q$

Let $\mathcal{P} = \{p_1, \ldots, p_m\}$ be given and fix a proposal $q$ such that $p_i \ll q$ for all $i$. Define for each $i$ the second-moment quantity

$$S_i(q) := \mathbb{E}_{z \sim q}\left[\left(\frac{p_i(z)}{q(z)}\right)^2\right] = 1 + \chi^2(p_i\|q).$$

Let $i^\star \in \arg\max_{i \in [m]} S_i(q)$ and write $w(z) := \frac{p_{i^\star}(z)}{q(z)}$ and $S := S_{i^\star}(q) = \mathbb{E}_q[w^2]$. Note that $S = 1 + \chi^2(p_{i^\star}\|q) \geq 1 + V(\mathcal{P})$ since $\max_i \chi^2(p_i\|q) \geq V(\mathcal{P})$ for every $q$.

We now construct two *stochastic* models $f^+, f^-$ whose induced losses under a query point $z$ are Bernoulli random variables with opposite biases. Fix $\varepsilon \in (0, 1/8)$ and define

$$h(z) := \frac{w(z)}{\sqrt{S}}, \qquad \alpha := \frac{\varepsilon}{\sqrt{S}}.$$

Assume the proposal $q$ satisfies the pointwise bound (as in Theorem 3.5)

$$\operatorname*{ess\,sup}_z w(z) \leq \frac{cS}{\varepsilon} \quad \text{for a sufficiently small universal constant } c \leq \tfrac{1}{8}.$$

Then for all $z$, $|\alpha h(z)| = \varepsilon w(z)/S \leq c \leq 1/8$.[5]

Define two stochastic models $f^+$ and $f^-$ by specifying their induced loss distributions: for each query point $z$, the observed loss is drawn independently as

$$\ell_{f^\pm}(z) \sim \mathrm{Bern}\big(\tfrac{1}{2} \pm \alpha h(z)\big).$$

---

[5]This property is crucial and ensures that the induced loss $\ell_{f^\pm}(z, u)$ follows a well-defined Bernoulli distribution.

**Risk separation under $p_{i^\star}$.** By construction,

$$\bar{\ell}_{f^+}(z) - \bar{\ell}_{f^-}(z) = 2\alpha h(z).$$

Therefore,

$$
\begin{aligned}
\mu_{i^\star}(f^+) - \mu_{i^\star}(f^-) &= \mathbb{E}_{z \sim p_{i^\star}}\big[\bar{\ell}_{f^+}(z) - \bar{\ell}_{f^-}(z)\big] \\
&= 2\alpha\, \mathbb{E}_{z \sim p_{i^\star}}[h(z)] \\
&= 2\alpha\, \mathbb{E}_{z \sim q}[w(z)h(z)] \\
&= 2\alpha \cdot \frac{\mathbb{E}_q[w^2]}{\sqrt{S}} \; = \; 2\varepsilon.
\end{aligned}
$$

### B.3. KL bound under $q$

Let $\mathbb{P}_+$ and $\mathbb{P}_-$ denote the distributions of the evaluator's transcript when the underlying model is $f^+$ or $f^-$, respectively. Since the protocol draws $Z_t \sim q$ i.i.d. and the losses are conditionally independent given $Z_t$, the KL divergence satisfies

$$\mathrm{KL}(\mathbb{P}_+\|\mathbb{P}_-) = n\, \mathbb{E}_{z \sim q}\big[\mathrm{KL}\big(\mathrm{Bern}(\tfrac{1}{2} + \alpha h(z)) \,\big\|\, \mathrm{Bern}(\tfrac{1}{2} - \alpha h(z))\big)\big].$$

For $|t| \leq 1/8$, a standard bound for Bernoulli KL (c.f. proof of Corollary A.2) gives

$$\mathrm{KL}\big(\mathrm{Bern}(\tfrac{1}{2} + t) \,\big\|\, \mathrm{Bern}(\tfrac{1}{2} - t)\big) \; \leq \; 32t^2. \tag{6}$$

Applying (6) with $t = \alpha h(z)$ and using $\mathbb{E}_q[h(z)^2] = 1$ yields

$$\mathrm{KL}(\mathbb{P}_+\|\mathbb{P}_-) \leq 32n\alpha^2 \mathbb{E}_q[h^2] = 32n \cdot \frac{\varepsilon^2}{S}.$$

Hence, if $n \leq \frac{S}{256\,\varepsilon^2}$, then $\mathrm{KL}(\mathbb{P}_+\|\mathbb{P}_-) \leq 1/8$ and by (5)

$$\mathrm{TV}(\mathbb{P}_+, \mathbb{P}_-) \leq \sqrt{\tfrac{1}{2} \cdot \tfrac{1}{8}} = \tfrac{1}{4}.$$

It follows that any test $\psi$ based on the transcript must have error at least $3/8$ under one of the two hypotheses:

$$\inf_\psi \max\{\Pr_+(\psi = -), \Pr_-(\psi = +)\} \; \geq \; \frac{1}{2}\big(1 - \mathrm{TV}(\mathbb{P}_+, \mathbb{P}_-)\big) \; \geq \; \frac{1}{2}\big(1 - \tfrac{1}{4}\big) = \frac{3}{8}.$$

In particular, no test can achieve error $\leq \delta$ for any fixed $\delta < 3/8$.

### B.4. Concluding the lower bound

Assume an evaluation protocol is $(\varepsilon, \delta)$-generalizable for the full risk profile, with $\delta < 3/8$, when run with i.i.d. samples from $q$. Applying this guarantee to environment $i^\star$ and models $f^+, f^-$, Lemma B.1 would yield a test with error at most $\delta < 3/8$, contradicting the preceding KL/TV bound whenever $n \leq \frac{S}{256\,\varepsilon^2}$. Therefore any such protocol must satisfy

$$n \; \geq \; \Omega\Big(\frac{S}{\varepsilon^2}\Big) \; = \; \Omega\Big(\frac{1 + \chi^2(p_{i^\star}\|q)}{\varepsilon^2}\Big).$$

Finally, since $i^\star$ maximizes $\chi^2(p_i\|q)$ and $\max_i \chi^2(p_i\|q) \geq V(\mathcal{P})$ for every $q$, we obtain

$$n \; \geq \; \Omega\Big(\frac{1 + V(\mathcal{P})}{\varepsilon^2}\Big),$$

which implies the stated $\Omega(V(\mathcal{P})/\varepsilon^2)$ lower bound up to universal constants. $\qquad\square$

## C. Proof of Proposition 3.7

*Proof.* We prove a reduction from uniform risk profile estimation to identifying one of $m$ hypotheses, and then lower bound the number of adaptive queries needed for identification via mutual information and Fano.

**Step 1: A half-set family with two combinatorial properties.** Fix $D := \lceil 256 \log m \rceil$ (so $D$ is even for $m$ large; otherwise increase by 1). Let $\mathcal{Z} = [D]$ and let $q$ be the uniform distribution on $[D]$. We will define $m$ subsets $A_1, \ldots, A_m \subset [D]$ each of size $D/2$ satisfying:

(C1) (*pairwise overlap control*) for all $j \neq k$,

$$\frac{3D}{16} \leq |A_j \cap A_k| \leq \frac{5D}{16};$$

(C2) (*column balance*) for all $z \in [D]$,

$$\frac{m}{3} \leq \big|\{j \in [m] : z \in A_j\}\big| \leq \frac{2m}{3}.$$

Such a family exists by the probabilistic method: choose $A_1, \ldots, A_m$ i.i.d. uniformly among subsets of $[D]$ of size $D/2$. Condition (C2) holds with probability at least $1 - 2De^{-m/72}$ by a Chernoff bound and a union bound over $z \in [D]$; condition (C1) holds with probability at least $1 - 2\binom{m}{2}e^{-D/64}$ by Hoeffding's inequality for hypergeometric random variables and a union bound over pairs $(j, k)$ (Wu et al., 2023, Theorem D.1). With our choice $D = \Theta(\log m)$, both failure probabilities are $< 1/2$ for all $m$ large enough, hence there exists a deterministic realization satisfying (C1)–(C2). Fix such a realization.

For each $j \in [m]$, define the environment $p_j$ as the uniform distribution on $A_j$:

$$p_j(z) := \frac{2}{D} \mathbf{1}\{z \in A_j\}.$$

Then $p_j \ll q$ and $\frac{p_j(z)}{q(z)} \in \{0, 2\}$ for all $z$, hence

$$\chi^2(p_j \| q) = \sum_{z=1}^{D} \frac{p_j(z)^2}{q(z)} - 1 = \sum_{z \in A_j} \frac{(2/D)^2}{1/D} - 1 = \frac{D}{2} \cdot \frac{4}{D} - 1 = 1.$$

Therefore $\max_j \chi^2(p_j \| q) \leq 1$, implying $V(\mathcal{P}) \leq 1$ for $\mathcal{P} = \{p_1, \ldots, p_m\}$.

**Step 2: A hard $m$-ary family of models.** Fix $\varepsilon \in (0, 1/8)$ and define, for each $j \in [m]$, a (randomized) model $f^{(j)}$ by specifying that on query $z \in [D]$ it outputs a Bernoulli loss

$$L \sim \mathrm{Bern}\!\left(\tfrac{1}{2} + \tfrac{\varepsilon}{2} v_j(z)\right), \qquad v_j(z) := 2\mathbf{1}\{z \in A_j\} - 1 \in \{+1, -1\}.$$

Let $\mu_k(f^{(j)}) := \mathbb{E}_{z \sim p_k}[\mathbb{E}(L \mid z)]$ denote the risk under environment $p_k$. Since $p_k$ is uniform on $A_k$ and $|A_k| = D/2$,

$$\mu_k(f^{(j)}) = \mathbb{E}_{z \sim p_k}\!\left[\tfrac{1}{2} + \tfrac{\varepsilon}{2} v_j(z)\right] = \tfrac{1}{2} + \tfrac{\varepsilon}{2} \mathbb{E}_{z \sim p_k}[v_j(z)]$$

$$= \tfrac{1}{2} + \tfrac{\varepsilon}{2}\left(\frac{2|A_j \cap A_k|}{|A_k|} - 1\right) = \tfrac{1}{2} + \tfrac{\varepsilon}{2}\left(\frac{4|A_j \cap A_k|}{D} - 1\right).$$

If $k = j$, then $|A_j \cap A_j| = D/2$ so $\mu_j(f^{(j)}) = \tfrac{1}{2} + \tfrac{\varepsilon}{2}$. If $k \neq j$, condition (C1) implies $|A_j \cap A_k| \in [3D/16, 5D/16]$, hence

$$\mathbb{E}_{z \sim p_k}[v_j(z)] \in \left[-\frac{1}{4}, \frac{1}{4}\right] \quad \Rightarrow \quad \mu_k(f^{(j)}) \in \left[\tfrac{1}{2} - \tfrac{\varepsilon}{8}, \; \tfrac{1}{2} + \tfrac{\varepsilon}{8}\right].$$

In particular, under model $f^{(j)}$, environment $j$ is separated from all others by a gap of at least $3\varepsilon/8$.

Therefore, if a protocol outputs estimates $\widehat{\mu}_1, \ldots, \widehat{\mu}_m$ satisfying $\max_{k \in [m]} |\widehat{\mu}_k - \mu_k(f^{(j)})| \leq \varepsilon/16$, then

$$\widehat{\mu}_j \geq \tfrac{1}{2} + \tfrac{\varepsilon}{2} - \tfrac{\varepsilon}{16} = \tfrac{1}{2} + \tfrac{7\varepsilon}{16}, \qquad \max_{k \neq j} \widehat{\mu}_k \leq \tfrac{1}{2} + \tfrac{\varepsilon}{8} + \tfrac{\varepsilon}{16} = \tfrac{1}{2} + \tfrac{3\varepsilon}{16},$$

so $\arg\max_k \widehat{\mu}_k = j$. Hence $(\varepsilon/16, \delta)$-generalizable estimation of the full profile implies identification of $j$ with error probability at most $\delta$.

**Step 3: Information gained per adaptive query is $O(\varepsilon^2)$.** Let $J$ be uniform on $[m]$, and suppose the protocol interacts with the unknown model $f^{(J)}$. At round $t$, the protocol chooses a query $Z_t \in [D]$ as a (possibly randomized) function of the past transcript $\mathcal{H}_{t-1} = (Z_1, L_1, \ldots, Z_{t-1}, L_{t-1})$ and then observes

$$L_t \sim \text{Bern}\Big(\tfrac{1}{2} + \tfrac{\varepsilon}{2} v_J(Z_t)\Big),$$

independently of the past given $(J, Z_t)$.

We bound $I(J; \mathcal{H}_n)$. By the chain rule of mutual information ([Polyanskiy & Wu](#), 2025, Theorem 3.7),

$$I(J; \mathcal{H}_n) = \sum_{t=1}^{n} I\big(J; (Z_t, L_t) \mid \mathcal{H}_{t-1}\big).$$

Given $\mathcal{H}_{t-1}$, the query $Z_t$ is determined by the protocol (and its internal randomness), so $I(J; Z_t \mid \mathcal{H}_{t-1}) = 0$ and thus (using chain rule again):

$$I\big(J; (Z_t, L_t) \mid \mathcal{H}_{t-1}\big) = I(J; L_t \mid Z_t, \mathcal{H}_{t-1}).$$

Moreover, conditioned on $Z_t = z$, the distribution of $L_t$ depends on $J$ only through the single bit $B := v_J(z) \in \{+1, -1\}$. Hence, for every history $h$, $L_t \perp\!\!\!\perp J \mid (B, Z_t = z, \mathcal{H}_{t-1} = h)$ and $B$ is a deterministic function of $J$. Therefore,

$$I(J; L_t \mid Z_t = z, \mathcal{H}_{t-1} = h) = I(B; L_t \mid Z_t = z, \mathcal{H}_{t-1} = h).$$

Let $\pi_{z,h} := \Pr(B = +1 \mid Z_t = z, \mathcal{H}_{t-1} = h)$. Then $L_t \mid (B = +1, Z_t = z, \mathcal{H}_{t-1} = h) \sim P_+$ and $L_t \mid (B = -1, Z_t = z, \mathcal{H}_{t-1} = h) \sim P_-$, where $P_+ = \text{Bern}(1/2 + \varepsilon/2)$ and $P_- = \text{Bern}(1/2 - \varepsilon/2)$, and the marginal law is $\bar{P}_{z,h} := \pi_{z,h} P_+ + (1 - \pi_{z,h}) P_-$. Therefore, by [Polyanskiy & Wu](#) (2025, Theorem 3.2(a)),

$$I(B; L_t \mid Z_t = z, \mathcal{H}_{t-1} = h) = \pi_{z,h} \text{KL}(P_+ \| \bar{P}_{z,h}) + (1 - \pi_{z,h}) \text{KL}(P_- \| \bar{P}_{z,h}).$$

Since $\varepsilon \leq 1/8$, $\bar{P}_{z,h}$ has mean in $[3/8, 5/8]$ for all $\pi_{z,h} \in [0, 1]$, and hence $q(1-q) \geq 3/16$ for $q$ equal to this mean. Using $\text{kl}(p\|q) \leq (p-q)^2/(q(1-q))$, we obtain the uniform bound

$$\text{KL}(P_\pm \| \bar{P}_{z,h}) \leq \frac{(\varepsilon/2)^2}{3/16} = \frac{4}{3}\varepsilon^2,$$

and consequently

$$I(J; L_t \mid Z_t = z, \mathcal{H}_{t-1} = h) \leq I(B; L_t \mid Z_t = z, \mathcal{H}_{t-1} = h) \leq \frac{4}{3}\varepsilon^2$$

for all $z$ and $h$. Averaging over $(Z_t, \mathcal{H}_{t-1})$ yields $I(J; L_t \mid Z_t, \mathcal{H}_{t-1}) \leq \frac{4}{3}\varepsilon^2$ for every $t$, and therefore

$$I(J; \mathcal{H}_n) = \sum_{t=1}^{n} I\big(J; (Z_t, L_t) \mid \mathcal{H}_{t-1}\big) = \sum_{t=1}^{n} I(J; L_t \mid Z_t, \mathcal{H}_{t-1}) \leq \frac{4}{3}n\varepsilon^2.$$

**Step 4: Fano's inequality.** Let $\widehat{J}$ be any estimator of $J$ based on $\mathcal{H}_n$. Fano's inequality gives

$$\Pr(\widehat{J} \neq J) \geq 1 - \frac{I(J; \mathcal{H}_n) + \log 2}{\log m} \geq 1 - \frac{4n\varepsilon^2 + \log 2}{\log m}.$$

If $n \leq c \, (\log m)/\varepsilon^2$ for a sufficiently small universal constant $c > 0$, then $\Pr(\widehat{J} \neq J) \geq 1/3$.

By Step 2, any $(\varepsilon/16, \delta)$-generalizable full-profile estimator with $\delta \leq 1/6$ would yield an identifier with error at most $1/6$, contradicting the above bound. Rescaling $\varepsilon$ by a constant completes the proof of $n = \Omega((\log m)/\varepsilon^2)$ for $(\varepsilon, \delta)$-generalizability (absorbing constant factors into $\Omega(\cdot)$). $\qquad\square$

## D. Adaptive Querying Lower Bound via a Null-Reference and Reverse KL

This appendix proves an adaptive analogue of Theorem [3.5](#) for fully adaptive evaluation protocols. The key idea is to compare each alternative model to a *null* reference model $f_0$ (rather than comparing the two alternatives directly) and then use a triangle inequality in total variation. Crucially, we bound the *reverse* divergences $\text{KL}(\mathbb{P}_0 \| \mathbb{P}_\pm)$, so all expectations are taken under the reference execution, which avoids having to control hypothesis-dependent query drift.

### D.1. Setup

Let $\mathcal{P} = \{p_1, \ldots, p_m\}$ be distributions on a common domain $\mathcal{Z}$. An evaluation protocol $\mathcal{E}$ proceeds for $n$ rounds. At round $t$, given the past transcript $H_{t-1} = \{(Z_s, L_s)\}_{s=1}^{t-1}$ and internal randomness, the protocol chooses a query point $Z_t \in \mathcal{Z}$ (possibly at random), then observes a loss $L_t = \ell_f(Z_t) \in \{0, 1\}$. The protocol may be fully adaptive, i.e. $Z_t$ may depend on the entire past transcript (including past losses).

We consider stochastic models $f$ defined by specifying the conditional distribution of the observed loss given the queried point: for each $z \in \mathcal{Z}$, $L \mid (Z = z) \sim \mathrm{Bern}(\bar{\ell}_f(z))$ with $\bar{\ell}_f(z) \in [0, 1]$, and losses are conditionally independent across rounds given the query points.

### D.2. Null reference proposal

Define the *null* model $f_0$ by $\bar{\ell}_{f_0}(z) \equiv 1/2$ for all $z$, so that under $f_0$ the observed losses are i.i.d. $\mathrm{Bern}(1/2)$ and independent of the queried points. Let $\mathbb{P}_0$ denote the distribution of the full transcript under $f_0$. For each $t$, let $q_t^0(\cdot \mid H_{t-1})$ be the (random) conditional distribution of $Z_t$ given the past transcript under $\mathbb{P}_0$. Define the averaged reference proposal

$$q_0(\cdot) \; := \; \frac{1}{n} \sum_{t=1}^{n} \mathbb{E}_{\mathbb{P}_0}\big[q_t^0(\cdot \mid H_{t-1})\big]. \tag{7}$$

For any measurable $g : \mathcal{Z} \to \mathbb{R}$,

$$\frac{1}{n} \sum_{t=1}^{n} \mathbb{E}_{\mathbb{P}_0}[g(Z_t)] \; = \; \mathbb{E}_{z \sim q_0}[g(z)]. \tag{8}$$

Define the second-moment parameter

$$S_{q_0}(\mathcal{P}) \; := \; \max_{i \in [m]} \mathbb{E}_{z \sim q_0}\left[\left(\frac{p_i(z)}{q_0(z)}\right)^2\right] \; = \; 1 + \max_{i \in [m]} \chi^2(p_i \| q_0).$$

Assume $p_i \ll q_0$ for all $i$, and that the likelihood-ratio bound holds:

$$\max_{i \in [m]} \; \operatorname*{ess\,sup}_{z \in \mathcal{Z}} \frac{p_i(z)}{q_0(z)} \; \leq \; \frac{c \, S_{q_0}(\mathcal{P})}{\varepsilon} \tag{9}$$

for a sufficiently small universal constant $c > 0$.

### D.3. Estimation to testing and TV–KL

We use the same estimation-to-testing reduction as Lemma B.1. We also use Pinsker's inequality:

$$\mathrm{TV}(P, Q) \leq \sqrt{\tfrac{1}{2} \mathrm{KL}(P \| Q)}. \tag{10}$$

### D.4. Hard instance

Let $i^\star \in \arg\max_i \mathbb{E}_{q_0}[(p_i/q_0)^2]$ and define $w(z) := \frac{p_{i^\star}(z)}{q_0(z)}$ and $S := \mathbb{E}_{q_0}[w^2] = S_{q_0}(\mathcal{P})$. Let

$$h(z) := \frac{w(z)}{\sqrt{S}}, \qquad \alpha := \frac{\varepsilon}{\sqrt{S}}.$$

By (9), for all $z$ we have $|\alpha h(z)| = \varepsilon\, w(z)/S \leq c \leq 1/8$ (for small enough $c$), so the following Bernoulli models are well-defined.

Define two stochastic models $f^+$ and $f^-$ by

$$\bar{\ell}_{f^\pm}(z) := \tfrac{1}{2} \pm \alpha h(z), \qquad L_t \mid (Z_t = z) \sim \mathrm{Bern}(\bar{\ell}_{f^\pm}(z)),$$

with losses conditionally independent across rounds given the queries. Let $\mathbb{P}_+$ and $\mathbb{P}_-$ denote the induced transcript distributions under $f^+$ and $f^-$, respectively.

**Risk separation.** By construction, $\bar{\ell}_{f^+}(z) - \bar{\ell}_{f^-}(z) = 2\alpha h(z)$, hence

$$
\begin{aligned}
\mu_{i^\star}(f^+) - \mu_{i^\star}(f^-) &= \mathbb{E}_{z \sim p_{i^\star}}[\bar{\ell}_{f^+}(z) - \bar{\ell}_{f^-}(z)] \\
&= 2\alpha \, \mathbb{E}_{z \sim p_{i^\star}}[h(z)] = 2\alpha \, \mathbb{E}_{z \sim q_0}[w(z)h(z)] \\
&= 2\alpha \cdot \frac{\mathbb{E}_{q_0}[w^2]}{\sqrt{S}} = 2\varepsilon.
\end{aligned}
$$

### D.5. Reverse-KL bounds to the null

We bound $\mathrm{KL}(\mathbb{P}_0\|\mathbb{P}_+)$ (and similarly for $\mathbb{P}_-$). By the chain rule for KL (Polyanskiy & Wu, 2025, Theorem 2.16),

$$
\mathrm{KL}(\mathbb{P}_0\|\mathbb{P}_+) = \sum_{t=1}^{n} \mathbb{E}_{\mathbb{P}_0}\Big[\mathrm{KL}\big(\mathcal{L}_0(L_t \mid H_{t-1}) \,\big\|\, \mathcal{L}_+(L_t \mid H_{t-1})\big)\Big], \tag{11}
$$

where $\mathcal{L}_0(L_t \mid H_{t-1})$ (resp. $\mathcal{L}_+(L_t \mid H_{t-1})$) denotes the conditional distribution of the observation $L_t$ at round $t$, given the past history $H_{t-1}$, under hypothesis 0 (resp. $+$). Note that, although the transcript includes both the query $Z_t$ and the observation $L_t$, the query-selection rule is *identical* under $\mathbb{P}_0$ and $\mathbb{P}_+$; consequently, the KL contributions associated with $Z_t$ cancel in the chain rule, leaving only the conditional divergence between the distributions of $L_t$.

Under $\mathbb{P}_0$, conditional on $H_{t-1}$ the protocol selects $Z_t$ according to some (random) distribution, but regardless of $Z_t$ we have $\mathcal{L}_0(L_t \mid H_{t-1}) = \mathrm{Bern}(1/2)$. Under $\mathbb{P}_+$ we have $\mathcal{L}_+(L_t \mid H_{t-1}, Z_t) = \mathrm{Bern}(1/2 + \alpha h(Z_t))$. Therefore

$$
\mathrm{KL}\big(\mathcal{L}_0(L_t \mid H_{t-1}) \,\big\|\, \mathcal{L}_+(L_t \mid H_{t-1})\big) = \mathbb{E}_{\mathbb{P}_0}\Big[\mathrm{KL}\big(\mathrm{Bern}(\tfrac{1}{2}) \,\big\|\, \mathrm{Bern}(\tfrac{1}{2} + \alpha h(Z_t))\big) \,\Big|\, H_{t-1}\Big].
$$

For $|u| \le 1/8$, standard Bernoulli KL bound gives $\mathrm{KL}(\mathrm{Bern}(\tfrac{1}{2})\|\mathrm{Bern}(\tfrac{1}{2} + u)) \le 16u^2$. Applying this with $u := \alpha h(Z_t)$ (which satisfies $|u| \le 1/8$ by construction) and substituting into (11) yields

$$
\mathrm{KL}(\mathbb{P}_0\|\mathbb{P}_+) \le 16\alpha^2 \sum_{t=1}^{n} \mathbb{E}_{\mathbb{P}_0}[h(Z_t)^2]. \tag{12}
$$

Applying (8) with $g = h^2$ gives

$$
\frac{1}{n}\sum_{t=1}^{n} \mathbb{E}_{\mathbb{P}_0}[h(Z_t)^2] = \mathbb{E}_{z \sim q_0}[h(z)^2] = \frac{1}{S}\mathbb{E}_{q_0}[w^2] = 1,
$$

so $\sum_{t=1}^{n} \mathbb{E}_{\mathbb{P}_0}[h(Z_t)^2] = n$. Hence

$$
\mathrm{KL}(\mathbb{P}_0\|\mathbb{P}_+) \le 16n\alpha^2 = 16n\frac{\varepsilon^2}{S}. \tag{13}
$$

The same argument gives $\mathrm{KL}(\mathbb{P}_0\|\mathbb{P}_-) \le 16n\,\varepsilon^2/S$.

### D.6. From reverse KL to indistinguishability

By Pinsker (10),

$$
\mathrm{TV}(\mathbb{P}_0, \mathbb{P}_\pm) \le \sqrt{\tfrac{1}{2}\mathrm{KL}(\mathbb{P}_0\|\mathbb{P}_\pm)} \le \sqrt{8n\frac{\varepsilon^2}{S}}.
$$

By the triangle inequality in total variation,

$$
\mathrm{TV}(\mathbb{P}_+, \mathbb{P}_-) \le \mathrm{TV}(\mathbb{P}_+, \mathbb{P}_0) + \mathrm{TV}(\mathbb{P}_-, \mathbb{P}_0) \le 2\sqrt{8n\frac{\varepsilon^2}{S}}.
$$

In particular, if $n \le S/(512\,\varepsilon^2)$ then $\mathrm{TV}(\mathbb{P}_+, \mathbb{P}_-) \le 1/4$, and therefore any test $\psi$ based on the transcript must incur error at least $3/8$ under one of the two hypotheses:

$$
\inf_{\psi}\max\{\Pr_+(\psi = -), \Pr_-(\psi = +)\} \ge \tfrac{1}{2}\big(1 - \mathrm{TV}(\mathbb{P}_+, \mathbb{P}_-)\big) \ge \tfrac{3}{8}.
$$

### D.7. Concluding the adaptive lower bound

Assume an evaluation protocol is $(\varepsilon, \delta)$-generalizable with $\delta < 3/8$. Applying this guarantee to environment $i^\star$ and models $f^+, f^-$, Lemma B.1 yields a test with error at most $\delta$, contradicting the above testing lower bound whenever $n \leq S/(512\,\varepsilon^2)$. Therefore any such protocol must satisfy

$$n = \Omega\left(\frac{S}{\varepsilon^2}\right) = \Omega\left(\frac{S_{q_0}(\mathcal{P})}{\varepsilon^2}\right).$$

Since $S_{q_0}(\mathcal{P}) \geq 1 + V(\mathcal{P})$, this also implies $n = \Omega(V(\mathcal{P})/\varepsilon^2)$ up to universal constants. $\qquad\square$

