# OpenReview forum: "Certifying Capabilities from Finite Tests: When Is It Possible?"
_ICML.cc/2026/Conference — ICML 2026 spotlight_

### Official Review · Reviewer_7S8W · 2026-03-12

**Soundness:** 4
**Presentation:** 3
**Significance:** 3
**Originality:** 4
**Overall Recommendation:** 5
**Confidence:** 3

**Summary:**

Authors develop a formal theory of evaluation as inference over a task family. They investigate when a capability can be accurately inferred from making finitely many black-box queries. They analyze two regimes: stochastic multi-environment evaluation and worst-case rule-like capabilities. Authors show that near-optimal evaluation exists when a specific structure can be assumed for the former, while deriving impossibility results for the latter.

**Compliance With Llm Reviewing Policy:**

Affirmed.

**Final Justification:**

I have decided to maintain my score as 5 for the following reasons:

* Authors have adequately addressed the readability concerns raised by me and other reviewers in their rebuttals.
* My initial assessment of the strengths for each dimension that I identified in my initial review remains unchanged.
* Authors did address the main question that could have potentially raised my score, although, as I expected, their answer matched my initial assessment, corresponding to my initial overall score of 5.

**Key Questions For Authors:**

* **Q1**: How impactful do you expect your results to be for designing audits for frontier AI models? I expect that the impact might be somewhat limited and indirect, but sufficiently compelling paths to impact would be a good argument for increasing my overall rating.

**Limitations:**

yes

**Strengths And Weaknesses:**

**Soundness**

Strengths:
* The claims are supported well by theoretical analysis with proofs and reasonable assumptions. The analysis is thorough and careful, with attention paid to technical nuances, for example, the consideration of circuit realizability on line 438.


**Presentation**

Strengths:
* The writing is generally very clear, e.g. in section 2, the problem formulation is stated in a way that is precise and information-dense while maintaining readability.
* The paper is positioned clearly in the context of prior literature.

Weaknesses:
* **W1**: The stochastic task families subsection in the introduction might be too dense. Perhaps it could be improved by elaborating a bit more on motivation and intuition.
* **W2**: Conclusion and discussion are currently in the appendix. The paper would be more readable if these were included in the main paper.
* Nitpicks: Page 12 appears to be missing. On line 80 “ when |P|=m” -> “, where |P|=m”?


**Significance**

Strengths:
* The setting of the capability evaluation is relevant and important. The paper develops a theoretical framework and highlights theoretical limitations, which could inform evaluation design. Authors mention that impossibility results analogous to theirs could be applicable to neural networks and Transformers, and indeed, it seems that the circuit construction methods for the backdoor designs of Goldwasser et al. (2022) and Draguns et al. (2024) would enable such an extension of these results.


**Originality**

Strengths:
* The framework is novel, and the authors introduce a novel formulation that uses χ^2-radius as a complexity parameter for multi-environment evaluation. The impossibility results are adjacent to existing work, but the framing and the specific setting are novel, and the authors clearly describe the comparison to these existing results.

---

> ### Author Rebuttal · Authors · 2026-03-27
>
> We thank the reviewer for the thoughtful review and for the positive assessment of our paper’s contributions.
>
> **W1:** We agree that the stochastic task family subsection could benefit from additional motivation and intuition, and in the revision we will make this part more accessible by adding more high-level explanation before the formal development (inclucding e.g., the shifting deployment disitrbution interpretation as provided in the rebuttal of Reviewer wYwW).
>
> **W2:** We thank the reviewer for this suggestion. In the revision, we will move the conclusion and discussion into the main body (using the additional page), and improve readability.
>
> **Q1:** We expect the impact of our framework on frontier-AI evaluation to be primarily *foundational* rather than direct, but we believe it is still meaningful. Our main contribution is to provide a rigorous framework for reasoning about what an evaluation based on finitely many tests can and cannot justify. In particular, the paper identifies when finite evaluation can support statistically valid extrapolation to a larger task family, and when such extrapolation is impossible without additional structure. We believe this is relevant since it helps clarify what assumptions are needed for evaluation claims to be credible, how one should define the target deployment family, and how finite tests should be interpreted. More broadly, our framework also suggests viewing evaluation as an inference problem, or even as an adaptive query-design problem, which could help inform more principled evaluation design in realistic settings.
>
> **Typos:** We thank the reviewer for catching this issue. We will fix them in the revision.

---

> > ### Author Rebuttal · Reviewer_7S8W · 2026-04-02
> >
> > Authors have addressed the readability concerns raised by me and other reviewers, e.g. Q1aG, and I believe that the issues will be fixed in the revision. Regarding the questions about the impact of this work raised by me and wYwW, the rebuttals roughly align with my prior moderately positive expectations, so I will maintain my score as 5.

---

### Official Review · Reviewer_wYwW · 2026-03-15

**Soundness:** 3
**Presentation:** 3
**Significance:** 2
**Originality:** 3
**Overall Recommendation:** 4
**Confidence:** 3

**Summary:**

This paper establishes a theoretical framework for capability evaluation. The authors show that finite evaluations can certify AI capabilities in probabilistic settings governed by a chi-squared radius, but face fundamental impossibility limits for worst-case, rule-based capabilities.

**Compliance With Llm Reviewing Policy:**

Affirmed.

**Key Questions For Authors:**

- As mentioned in the weaknesses, how do the theoretical results translate into concrete guidance for practitioners designing evaluation benchmarks for LLMs in real-world settings?
- Regarding the combinatorial impossibility results: Are there structural assumptions on the model class or task family beyond the parity example that would enable positive guarantees?

**Limitations:**

Yes

**Strengths And Weaknesses:**

Strengths:
- Clearly written and well structured paper
- Framing evaluation as an inference problem rather than simple benchmark scoring is a fresh and compelling direction
- Gives evaluation a solid theoretical foundation it previously lacked
- Delivers tight, near-optimal results with a clear measure of sample complexity for the stochastic setting
- Shows limitation in the combinatorial regime

Weaknesses:
- While the formalism is rigorous, the practical guidance for evaluation practitioners remains somewhat lacking. The theory clarifies when certification is possible, but offers limited help on how to design better benchmarks in realistic settings. It would be valuable if the authors could elaborate on how the theoretical results map to the real-world evaluation of LLMs.
- From the formality point of view, the conclusion and discussion section would be better placed in the main body rather than the appendix. It would be worth to move parts of proofs in the appendix to make room for that. For example, the parity function example in the combinatorial setup could be better highlghted in the main text.

---

> ### Author Rebuttal · Authors · 2026-03-27
>
> We thank the reviewer for the thoughtful review and for highlighting the paper’s main contributions.
>
> **W1:** We appreciate the reviewer’s point that the practical implications could be made more explicit. Our primary goal in this paper is to establish a rigorous framework that casts evaluation as an *inference problem*: given only finitely many tests, what can one validly extrapolate about performance over a larger task family? We believe this provides a principled foundation for reasoning about evaluation under different assumptions, and for comparing evaluation methods by the strength of the guarantees they support. We will make this motivation more explicit in the revision and briefly discuss implications for real-world LLM evaluation (see **Q1** below).
>
> **W2:** We thank the reviewer for the suggestion. In the revision, we will move the conclusion into the main body (using the additional page) and make the parity-function example more prominent.
>
> **Q1:** As we mention in the discussion section, one concrete piece of guidance for practitioners is that benchmarks should be designed with a *specific extrapolation target* (i.e., the task family in our framework) in mind, rather than as static collections of test cases. Our results also motivate careful modeling of the extrapolation target itself.
>
> Operationally, one way to interpret our *multi-environment setting* is as follows: suppose we have access to a canonical task distribution (e.g., $P_0$), but the true deployment distribution may shift (for example, within a Wasserstein ball around $P_0$). Then the task family corresponds to the collection of all such shifted distributions. Our results provide an optimal way to estimate the model’s performance over this entire shifted family from finite queries.
>
> More broadly, our framework suggests that evaluation can benefit from being viewed as an active learning/estimation problem, where one selects informative tests adaptively rather than relying only on a fixed benchmark.
>
> **Q2:** Indeed, additional structural assumptions on the model class and/or task family can enable positive guarantees in the combinatorial setting. Beyond the parity example, a natural case is the class of Boolean $k$-juntas, where the function depends on only $k$ underlying variables. In this case, uniform random sampling can distinguish perfect performance from nontrivial error using $O(2^k \log(1/\delta))$ queries, which is polynomial when $k \le \log n$. Extending such examples to a more general framework is an interesting direction for future work.

---

> > ### Author Rebuttal · Reviewer_wYwW · 2026-04-05
> >
> > Thanks for the rebuttal. I keep my score.

---

### Official Review · Reviewer_Q1aG · 2026-03-15

**Soundness:** 3
**Presentation:** 1
**Significance:** 3
**Originality:** 3
**Overall Recommendation:** 4
**Confidence:** 3

**Summary:**

This is a theory paper about certifying capabilities of large language models (LLMs) using finite tests.  In this direction the authors formalize tasks and capabilities (risk profiles) of LLMs and investigate the extent to which guarantees can be inferred over a task family by running tests on a finite subset of possibilities.  The authors provide different results for different kinds of task families (stochastic vs combinatorial).  Regarding stochastic task families the authors provide positive results for certification via finite tests.  The idea is that since the tasks are stochastic, by performing a sufficiently large number of tests, essentially generalization can be accomplished from iid sampling.  On the other hand, for combinatorial tasks (e.g., addition of k-digit numbers) one requires correctness over the entire set of possible instances that need to be answered and this time the authors provide a result that roughly says that no certification can be accomplished using a sub-exponential amount of queries.

**Compliance With Llm Reviewing Policy:**

Affirmed.

**Final Justification:**

I trust that the authors will try to improve the presentation in the final version of the paper.  However, given that I think the presentation could improve a lot, I would like to maintain my score of weak accept.

**Key Questions For Authors:**

No questions.

**Limitations:**

yes

**Strengths And Weaknesses:**

The paper appears to be sound.  To the extent that I checked the ideas and the proofs, they appear to be correct.  However, I believe that the presentation can be improved.  I think the paper would benefit with a section of preliminaries where notation is defined and additional background information can be provided for the tools that are being used.  For example, Bern in line 212 (right column) I do not believe that is defined.  As another example, while Chebyshev's inequality is a standard tool for providing results in the spirit of the paper, nevertheless, I think the authors could mention it as a reminder to the readers in the preliminaries section. Additional information on $\chi^2$, on the KL-divergence and its use and why it is appropriate for their methods rather than other approaches or even the symmetric KL-divergence, Yao's minimax principle, etc.  Other than that, I do not see major issues, but I do see this as a major drawback/limitation of the paper.

---

> ### Author Rebuttal · Authors · 2026-03-27
>
> We thank the reviewer for the positive assessment of our paper’s contributions. We also agree that the presentation can be improved.
> In the revision, we will add a clearer preliminaries/background section to define notation systematically (including $\mathrm{Bern}(\cdot)$) and briefly recall the main tools used in the paper, with precise citations where appropriate, such as Chebyshev’s inequality, divergence-based quantities, and Yao’s minimax principle. We will also add more intuition and roadmap text to better connect the abstract framework to the stochastic and combinatorial results.

---

> > ### Author Rebuttal · Reviewer_Q1aG · 2026-04-02
> >
> > Thank you for the response. Given that I had some hard time with the presentation in the paper I would like to maintain my score.

---

### Official Review · Reviewer_fyjm · 2026-03-19

**Soundness:** 4
**Presentation:** 3
**Significance:** 4
**Originality:** 4
**Overall Recommendation:** 5
**Confidence:** 3

**Summary:**

Typically, the capability of a model (or models) is assessed through an evaluator. However, any evaluator can only test the model's performance on finitely many “tasks.” What does this say about the model’s performance across all tasks? Just based on these evaluations, we may not be able to infer much about performance beyond the evaluated tasks.  Can we design evaluators that will test the model on finitely many instances and yet can infer beyond what is tested? This work is an attempt to provide a general theoretical framework to address this question.

The authors define a general framework by defining tasks and evaluators . Then they instantiate this framework in two concrete settings: a stochastic setting and a combinatorial setting. In the stochastic setting, the authors provide upper and lower bounds on the sample complexity of an evaluator, relating these bounds to the $\chi^2$-divergence of the distributions. In the combinatorial setting, the authors obtain general impossibility results.

**Compliance With Llm Reviewing Policy:**

Affirmed.

**Key Questions For Authors:**

Please see W2 and W3.

Additionally,
1. Line 207 (first column): \Delta is undefined.
2. Line 202 (second column: What does it mean by "amost surely"?
3. Equation 1, What does p_i << q mean?

**Limitations:**

Yes

**Strengths And Weaknesses:**

Strengths:


1. The attempt to provide a general theoretical framework for evaluating model capabilities is valuable, timely, and is well-motivated.
2. The upper and lower bounds are clean and interesting.
3. To obtain the lower bound of m/\epsilon^2, the authors reduce the problem to estimating bias of m coins, which is neat.
4. The use of \chi^2 divergence in both upper and lower bounds is elegant and interesting
5. The paper is a starting point in this direction, and could potentially lead to further future work.

Weakness:

1. I found Section 2 difficult to follow. I started to appreciate the paper more after reading Section 3. While I understand that the authors are trying to provide a general framework, the abstraction  in Section 2 makes it harder to understand.

2.  There are some inconsistencies in how the general model is defined. For example, in lines 151–157 (first column), $L_t$ is defined only for tasks in family $\Tau$. But Definition 2.2 says that q_k in general may not lie in the task family. How does an evaluator know the loss in such cases? This is critical in the proof of Theorem 3.4, the distribution q is not among the tasks. As far as I can see, the framework seems to implicitly assume a larger query/task space. This is not stated clearly (or I am misunderstanding).

3.  In the proposed definition, the evaluator is supposed to work well against "all" models. This is crucially exploited in lower bound proofs, for example  Proposition 3: Lowerbound is obtained by carefully defining  "adversarial" models that mimic coin tosses. In practice, reasonable models are not adversarial and do not behave like this. Perhaps an alternate definition could be to fix a finite set of "known" models and ask that an evaluator work on those models only. What can we say about this?

---

> ### Author Rebuttal · Authors · 2026-03-27
>
> We thank the reviewer for the detailed review and helpful comments. We now address the main questions raised below.
>
> **W1:** We thank the reviewer for this helpful comment. We agree that Section 2 is relatively abstract, and that its role is to provide a general framework that is instantiated more concretely in Section 3. In the revision, we will improve the presentation of Section 2 by clarifying the high-level intuition, more explicitly separating the deployment task family from the evaluator’s query space, and adding earlier pointers to the concrete multi-environment setting in Section 3.
>
> **W2:** We thank the reviewer for catching this discrepancy. The reviewer is correct that our evaluation model implicitly allows a larger *query space* than the deployment task family, and we will make this explicit in the revision. Concretely, the role of the task family $\mathcal{P}$ is to *define* the risks/capabilities of interest via the risk distributions $\mathcal{L}_t$ in Definition 2.1. By contrast, the evaluator is allowed to issue auxiliary queries that need not themselves belong to $\mathcal{P}$. The only requirement is that, for any allowed query $(x,y)$ to model $f$, the evaluator can observe an induced loss (depending on the specific access model).
>
> In the multi-environment setting of Section 3 (the setting underlying Theorem 3.4), this access model is instantiated as follows: we assume a known loss function $\ell$ such that any query $(x,y)$ to model $f$ returns the loss $\ell(f(x),y)$. Therefore, even when the query distribution $q$ is not itself a task in $\mathcal{P}$, the evaluator still observes well-defined loss feedback through $\ell(f(x),y)$. Crucially, for $(x,y)$ outside the support of the deployment task family, this loss *need not* have an independent semantic interpretation as a “task”; it is used only as an observable signal for statistical inference.
>
> Indeed, the only assumption needed by Theorem 3.4 is that the task risks and the query risks use the *same* underlying loss $\ell$. Under this assumption, the theorem shows that one can construct a query distribution $q$ from the task family $\mathcal{P}$ such that the *actual* risk profile of $\mathcal{P}$ can be statistically inferred from the losses induced by samples from $q$, even though $q$ itself may lie outside the task family and its samples may not have the same semantic meaning as deployment tasks.
>
> **W3:** We would like to clarify that the notion of “tasks” (i.e., the “models” referred to by the reviewer) is intended to capture the *actual* deployment distributions that the system (e.g., an LLM) may face in the real world. Accordingly, the goal of evaluation is to choose a finite set of tests, via queries to the system, that can statistically *certify* the system’s behavior when deployed on these tasks.
> In this paper, we study the most pessimistic setting, namely a worst-case/adversarial formulation, in which we seek the *strongest* certificates that remain valid for the *worst-case* task in the task family.
>
> We agree with the reviewer that one could also consider a weaker alternative in which the evaluator is only required to work for a fixed finite class of known tasks. That restricted setting is certainly interesting for future investigation, and we expect the evaluation problem to become easier there. More broadly, the choice of task family ultimately reflects how strong one wants the resulting certificate to be.
>
> **Other questions:**
> - For any given set $S$, $\Delta(S)$ denotes the set of all probability distributions over $S$.
> - Here, “almost surely” means that $(X,Y) = (x_i,0)$ holds with probability 1.
> - “$p_i \ll q$” means that $p_i$ is absolutely continuous with respect to $q$; in other words, any event with zero probability under $q$ must also have zero probability under $p_i$.

---

> > ### Author Rebuttal · Reviewer_fyjm · 2026-04-01
> >
> > Thank you for the responses. I am keeping my score as it is.

---

### Decision · Program_Chairs · 2026-04-30

**Decision:**

Accept (spotlight)

**Comment:**

This paper focuses on a timely topic and the reviewers appreciated the contribution. The authors should take into account feedback for the final version of the paper.